# Challenges in replay detection by TDLM in post-encoding resting state

Simon Kern[1,2,3,4]*, Juliane Nagel[1,2,3,4], Lennart Wittkuhn[5], Steffen Gais[6], Raymond J Dolan[7,8], Gordon B Feld[1,2,3,4]

[1]Clinical Psychology, Central Institute of Mental Health, Medical Faculty Mannheim, University of Heidelberg, Mannheim, Germany; [2]Psychiatry and Psychotherapy, Central Institute of Mental Health, Medical Faculty Mannheim, University of Heidelberg, Mannheim, Germany; [3]Addiction Behavior and Addiction Medicine, Central Institute of Mental Health, Medical Faculty Mannheim, University of Heidelberg, Mannheim, Germany; [4]Department of Psychology, Ruprecht Karl University of Heidelberg, Heidelberg, Germany; [5]Institute of Psychology, Universität Hamburg, Hamburg, Germany; [6]Institute of Medical Psychology and Behavioral Neurobiology, Eberhard-Karls-University Tübingen, Tübingen, Germany; [7]Max Planck UCL Centre for Computational Psychiatry and Ageing Research, London, United Kingdom; [8]Wellcome Centre for Human Neuroimaging, University College London, London, United Kingdom

*For correspondence: simon.kern@zi-mannheim.de

## eLife Assessment

This study presents **valuable** findings on the ability of a state-of-the-art method, Temporally Delayed Linear Modelling (TDLM), to detect the replay of sequences in human memory. The investigation provides **compelling** evidence that TDLM has significant limitations in its sensitivity to detect replay in extended (minutes-long) rest periods. The work will be of strong interest to researchers investigating memory reactivation in humans, especially using iEEG, MEG, and EEG.

**Abstract** Using temporally delayed linear modeling (TDLM) and magnetoencephalography (MEG), we investigated whether items associated with an underlying graph structure are replayed during a post-learning resting state. In these same data, we previously provided evidence for replay during online (non-rest) memory retrieval. Despite successful decoding of brain activity during a localizer task, and contrary to predictions, we found no evidence for replay during a post-learning resting state. To better understand this, we performed a hybrid simulation analysis in which we inserted synthetic replay events into a control resting state recorded prior to the actual experiment. This simulation revealed that replay detection using our current pipeline requires an extremely high replay density to reach significance (>1 replay sequence per second, with 'replay' defined as a sequence of reactivations within a certain time lag). Furthermore, when scaling the number of replay events with a behavioral measure, we were unable to induce a strong correlation between sequenceness and this measure. We infer that even if replay was present at plausible rates in our resting state dataset, we would lack statistical power to detect it with TDLM. Finally, contrasting our novel hybrid simulation to existing purely synthetic simulations indicated that the latter approaches overestimate the sensitivity of TDLM. We discuss approaches that might optimize the analytic methodology, including identifying boundary conditions under which TDLM can be expected to detect replay. We conclude that solving these methodological constraints will be crucial for optimizing the non-invasive measurement of human replay using MEG.

## Introduction

Reactivation of memory traces during offline states is considered a core mechanism supporting memory consolidation. Such memory reactivation has been investigated primarily in animal models, particularly in the context of hippocampal place cells (*O'Keefe, 1976*), wherein firing patterns that encode movements through a maze re-emerge later during rest periods (*Pavlides and Winson, 1989*), with their occurrence correlating with post-rest performance. Subsequently, memory reactivation has been linked to consolidation, that is the stabilization of long-term memory traces (*Born and Wilhelm, 2012*; *Feld and Born, 2017*; *Feld and Born, 2020*; *Klinzing et al., 2019*; *Diekelmann and Born, 2010*; *Findlay et al., 2020*; *Foster, 2017*; *Wilson and McNaughton, 1994*). Here, 'memory reactivation' refers to the re-emergence of activation patterns absent an eliciting sensory stimulus (*Genzel et al., 2020*). 'Memory replay', on the other hand, emphasizes a sequential ordering of reactivations, where replay patterns follow a forward or backward trajectory (*Diba and Buzsáki, 2007*; *Foster and Wilson, 2006*). For example, backward replay is linked to reward processing during breaks in task performance (*Ambrose et al., 2016*). An additional feature of replay is its time compression, such that its time course is much faster than the temporal evolution of the original encoded activity (*Nádasdy et al., 1999*).

Replay has primarily been demonstrated using cellular recordings within a large variety of mammalian model organisms (*Hoffman and McNaughton, 2002*; *Lee and Wilson, 2002*; *Pavlides and Winson, 1989*). Replay studies in humans using intracranial microelectrode recordings are few, but include work demonstrating compressed replay of firing-pattern sequences in motor cortex during rest and sleep (*Eichenlaub et al., 2020*; *Rubin et al., 2022*) as well as single-unit replay of trial-specific cortical spiking sequences during episodic retrieval (*Vaz et al., 2020*). By contrast, studies using macroelectrodes mainly report stimulus-specific reinstatement or ripple-locked activity changes, as a correlate of replay (*Zhang et al., 2025*), without explicit demonstration of temporal sequential cellular replay (e.g. *Axmacher et al., 2008*; *Staresina et al., 2015*; *Zhang et al., 2018*). As these methods can only be applied under restricted clinical circumstances, such as during pre-operative neurosurgical assessments, this limits opportunities to investigate human replay. Therefore, this gives urgency to efforts aimed at developing novel methods to investigate human replay non-invasively.

Non-invasive neuroimaging techniques suffer a trade-off (*Hall et al., 2014*) – either their spatial precision is high but their temporal precision is lacking, as with functional magnetic resonance imaging (fMRI), or vice versa as with magneto- or electroencephalography (MEG/EEG). This has motivated the development of sophisticated analytical approaches to overcome these limitations. While measuring faster sequences is possible with fMRI (*Schuck and Niv, 2019*; *Wittkuhn et al., 2024*; *Wittkuhn and Schuck, 2021*), most recent human replay research has focused on MEG- or EEG-based analysis (*Liu et al., 2022b*).

Using advances in brain decoding that also exploit the temporal precision of MEG, an emerging approach to quantifying replay involves examining lag differences between individually decoded reactivation events (*Kurth-Nelson et al., 2016*; *Liu et al., 2021a*), a method referred to as 'temporally delayed linear modeling' (TDLM). In essence, by computing a 'sequenceness' score, the method quantifies whether a predicted sequence of decoded brain activity appears more often than expected by chance. Using this approach, a series of studies have provided evidence that individual states are replayed in sequence not only during rest (*Eldar et al., 2020*; *Huang et al., 2024*; *Liu et al., 2019*; *Nour et al., 2021*; *Nour et al., 2023*; *Yu et al., 2023*) but also in the context of active cognition including planning/decision making (*Eldar et al., 2020*; *Huang et al., 2020*; *Kurth-Nelson et al., 2016*; *McFadyen et al., 2023*; *Schwartenbeck et al., 2023*; *Seow et al., 2026*; *Wimmer et al., 2023*; *Wise et al., 2021*), memory retrieval (*Huang and Luo, 2023*; *Kern et al., 2024*; *Wimmer et al., 2020*) and following receipt of reward (*Liu et al., 2021b*).

Relevant to the current study, off-line replay of task-relevant sequences has been reported across several studies. *Liu et al., 2019* reported on participants who first learned a rule-based mapping of visually presented items, followed by a 5-min recording of resting state activity, prior to completion of a subsequent inference task. In one study iteration, value learning was applied to one of two sequences of sensory items. During the resting session, states were replayed with an approximate 40–60 ms state to state lag, in a predominantly forward or backward direction, depending on whether a specific sequence was rewarded or not. *Eldar et al., 2020* found sequenceness during resting state was associated with lessened flexibility to subsequent changes in value-location pairings, consistent

with replay supporting a form of model-free decision-making. In the latter, backward replay was reported during rest at a 180 ms time lag, albeit using a cross-correlation approach, a predecessor method to TDLM. *Nour et al., 2021* found attenuated rest period replay at a time lag of 40–50 ms in patients with schizophrenia compared to healthy controls. In the same data, *Nour et al., 2023* found that patients showed reduced coupling of replay to default-mode-network activation, an impairment that correlated with memory retention. *Yu et al., 2023* used EEG recordings in a sequence-learning task and found that participants with high trait anxiety showed reduced reverse replay of reward sequences during a resting state. Finally, *Huang et al., 2024*, using simultaneous EEG-fMRI to examine the spatial and temporal dynamics of memory replay in humans, showed that during mental simulation, replay events were associated with activation in the hippocampus and medial prefrontal cortex.

In the current study, our focus was on the consolidation of graph-based task structures during resting state as captured in MEG data. Previous research indicates that recently acquired mnemonic memories are replayed in subsequent rest or sleep sessions (*Humiston et al., 2019*; *Schapiro et al., 2018*; *Schreiner et al., 2021*; *Tambini et al., 2010*; *Wamsley, 2022*), with replay density decreasing over a time window of 2 hr (*Eschenko et al., 2008*). Additionally, there is evidence that replay at rest, and during sleep, preferentially consolidates items based on properties such as encoded strength (*Drosopoulos et al., 2007*; *Huelin Gorriz et al., 2023*; *Schapiro et al., 2018*) or graph node properties (*Feld et al., 2022*). Based on this, in our study, we employed criterion learning to 80% correct retrieval on a graph network, a criterion similar to the performance participants achieved on a similar graph-based task which showed a behavioral sleep benefit (*Feld et al., 2022*). We assumed this would achieve a level of memory acquisition that engenders subsequent sequence replay, as it allowed for a further memory improvement, which we conjectured depends on replay. Such criterion learning is standard practice in studies on sleep-dependent memory consolidation (e.g. *Feld et al., 2013*; *Galer et al., 2015*; *Payne et al., 2012*; *Rasch et al., 2006*).

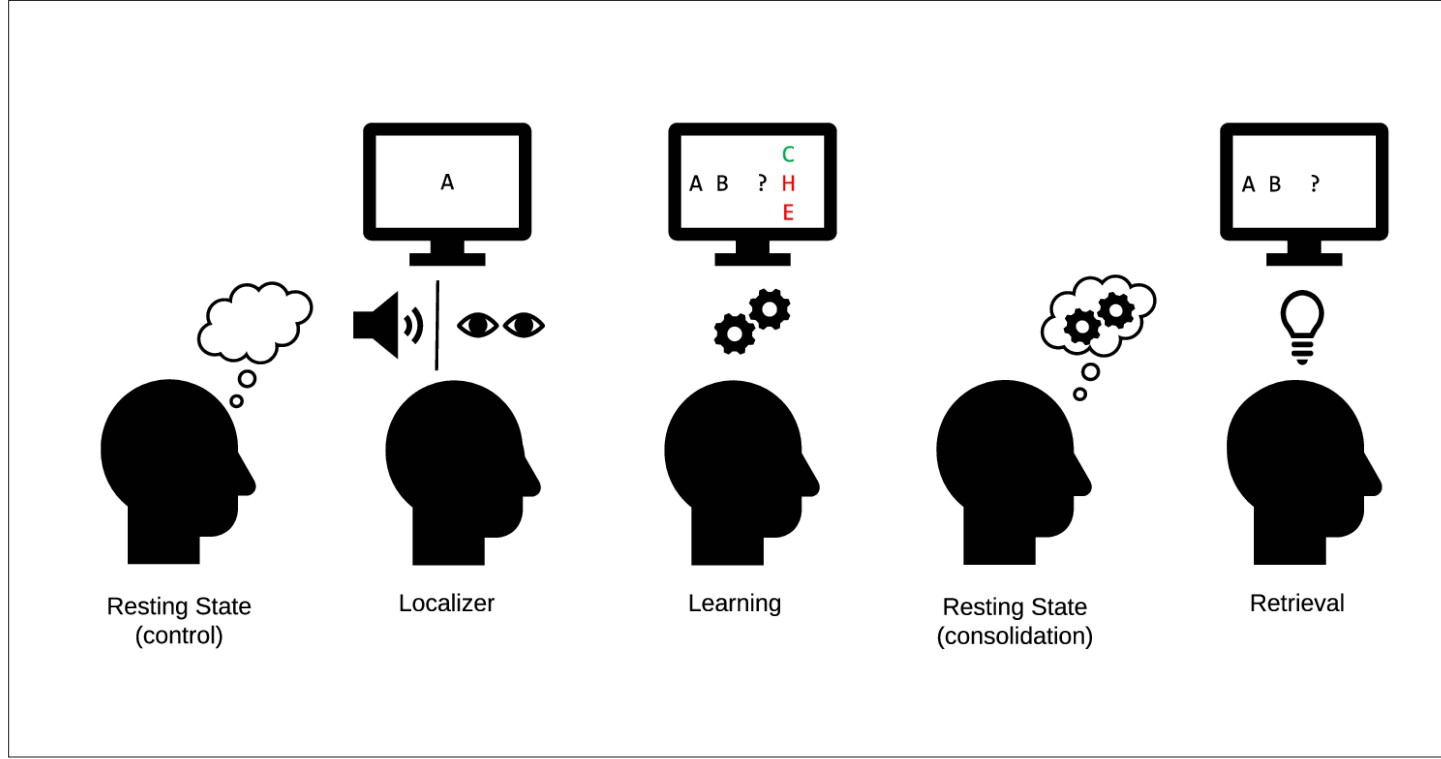

**Figure 1.** Experimental procedure in the MEG. Control: 8 min of resting state activity were recorded before participants encountered any stimuli. Localizer task: Next, the 10 individual items were repeatedly presented to participants auditorily and visually to extract multisensory activity patterns. Learning: Participants learned triplets of the ten items pseudo-randomly generated based on an underlying graph, by performing a three-alternatives-forced choice. Participants were unaware of the graph layout. Consolidation: 8 min of resting state activity were recorded. Retrieval: Participants' recall was tested by cueing triplets from the graph. The letters in the pictograms are placeholders for individual images. Figure and caption adapted from *Kern et al., 2024*. For details, see Materials and methods section *Figure 10*. For an animated overview of the task and the experiment, see also our GitHub pages of the study: here.

Here, we acknowledge that previous studies using TDLM focused primarily on structural inference and decision making and not on mnemonic learning (*Eldar et al., 2020*; *Liu et al., 2019*; *Liu et al., 2021b*; *Nour et al., 2021*; *Nour et al., 2023*). Nevertheless, we considered it likely that there would also be replay after mnemonic learning (*Buhry et al., 2011*; *Findlay et al., 2020*; *Zhou et al., 2024*) and, if present, this would be amenable to quantification using TDLM. Despite this, we did not find replay evidence in our post-learning resting state session, leading us to conduct a second study wherein we simulated replay by inserting neural events taken from the localizer data, at varying densities, into a control resting state recorded before any stimulus exposure. This simulation allowed us to quantify a lower bound of replay density necessary to detect replay. Additionally, to simulate a previously demonstrated association of replay and performance (e.g. *Schapiro et al., 2018*; *Wimmer et al., 2020*), we scaled simulated densities by participant performance. Finally, we discuss existing uncertainties regarding parameter choices and possible resulting confounders when using TDLM as well as suggesting ways to improve the robustness of the method.

## Results

### Results Study I: resting state analysis

In Study I, participants were first placed in an MEG scanner for acquisition of data during an 8-min resting state (see *Figure 1* for an overview of the procedure). Next, while subjects remained in the scanner, we performed a localizer task on which a classifier was trained. Participants then learned triplets of associated items according to a graph structure. Within this learning session, participants performed a maximum of six learning blocks, but the session was stopped if participants reached 80% memory performance (criterion learning, see Materials and methods for details). Finally, participants were asked to remain in the scanner for a further 8-min resting state measurement, after which we tested memory retrieval. Mean learning performance of ~82% indicated that the applied criterion learning was successful.

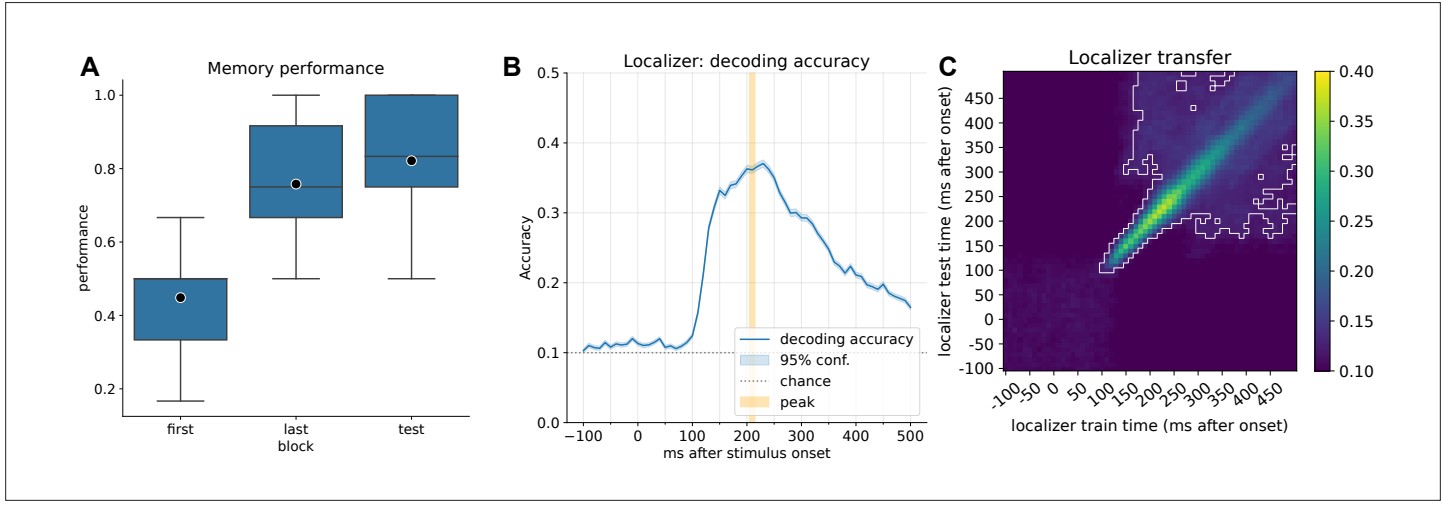

**Figure 2.** Behavioral results and localizer decoding accuracy. (**A**) Memory performance of participants after completing the first block of learning, the last block of learning (block 2–6, depending on speed of learning), and retrieval performance after the resting state. Participants completed up to six blocks of learning trials. After reaching 80% in any block, no more learning blocks were performed (criterion learning). (**B**) Decoding accuracy of the currently displayed item during the localizer task for participants with a decoding accuracy above 30% (n=21, 9 participants' data did not pass this criterion). The mean peak time point across all participants corresponded to 210 ms, with an average decoding peak accuracy of 42% (n=21). Note that the displayed graph combines accuracies across participants, where peak values were computed on an individual level and then averaged. Therefore, at a group level, the individual mean peak does not necessarily correspond to the average peak. (**C**) Classifier transfer within the localizer when trained and tested at different time points as determined by cross-validation, after ~200 ms, most training and testing time points generalize to each other (significant clusters indicated by cluster permutation testing, alpha <0.05). A visualization of the peak mean decoding accuracy across different regularization values, see *Figure 2—figure supplement 1*.

The online version of this article includes the following figure supplement(s) for figure 2:

**Figure supplement 1.** Cross-validation peak accuracy during the localizer across different regularization values.

In the post-task resting state, we found no evidence for sequenceness reflective of task structure. Behavioral results indicated that the experimental manipulation worked as intended, that is participants learned the graph to criterion, without overlearning which might have precluded the occurrence of replay (see *Figure 2*). Additionally, we tested for a correlation of memory performance and sequenceness strength, as group-level sequenceness thresholds might be too conservative alone, but found none. Note that data from the retrieval stage were published elsewhere (*Kern et al., 2024*). The study and analysis were preregistered at aspredicted.org (https://aspredicted.org/kx9xh.pdf, #68915).

## Behavioral results

All thirty participants, except one, learned the graph consisting of images with an accuracy above 50% (preregistered as an exclusion criterion for 'successful learning above chance'). After a consolidation resting state, a marginal increase in performance was observed (76–82% accuracy, t=−2.053, p=0.053, effect size *r*=0.22; *Figure 2A*), but note that this may not necessarily be attributed to consolidation, as the last block also included further feedback. More details on behavioral results are found in *Kern et al., 2024*.

## Decoder training

We trained classifiers on data from the localizer task, in which participants were presented with images from the learning task in a pseudorandom order. In each trial, a word describing the stimulus was played auditorily, after which the corresponding stimulus was shown. In ~11% of cases, there was a mismatch between word and image (oddball trials), and these trials were excluded from the localizer training. Decoding from brain activity was at a level comparable to previous reports as determined in cross-validation (~42% across all participants, range 32–57%, chance level: 10%, excluding participants with peak accuracy <30%). We used the average peak decoding accuracy time point of 210 ms across all participants (*Figure 2B*) to train final participant-specific decoders to apply on resting state data.

## No sequential replay during resting state

By applying trained decoders to the resting state session, we used probability estimates of the ten classes as input to TDLM. In brief, TDLM quantifies time-lagged reactivation ('sequenceness') of states according to expected transitions, for example, quantifying how strongly state A->B was reactivated with a temporal distance of N milliseconds. As there was no specific sub-hypothesis about which transitions would be replayed, we tested for all allowable graph transitions. We found no evidence for sequential replay measurable during the post-learning resting state. More specifically, the GLM did not exceed significance thresholds (conservative, i.e. the maximum sequenceness across all permutations and time points or liberal criteria, i.e. the 95% percentile of aforementioned sequenceness) for any individual time lag, neither in forward nor backward direction (*Figure 3A*).

Additionally, we calculated the difference between the pre-resting state and post-resting state sequenceness values to remove potential systemic biases on sequenceness fluctuation present in both sessions. We calculated an FDR-corrected paired t-test for all 30 time lags individually, for forward and backward sequenceness between pre- and post-resting state respectively. All time lags were non-significant ($p_{FDR} > 0.05$; without correction, three time points were significant: forward 100 ms, t=−2.43, p=0.024 and 240 ms, t=2.13, p=0.045, backward 200 ms time lag, t=−2.11, p=0.047). However, as noted in *Liu et al., 2021a*, correct statistical analysis of sequenceness scores is non-trivial and FDR-correction might be too conservative. The permutation test shuffles state labels and measures a fixed effect, making it susceptible to outliers on a group level, which we addressed by z-scoring the sequenceness results. We also included a newly developed method to take random effects at a group level into account, specifically employing a sign-flip permutation test that randomly inverts a subset of participants' sequences 10,000 times and calculates a one-sided t-test against null for each time lag per permutation. The maximum t-value per permutation is used to construct an empirical null distribution. This approach circumvents the multiple comparison problem inherent in the previous method (see Materials and methods section for details). Neither the control nor the post-learning resting state surpassed sign-flip significance thresholds (*Figure 3C* control: forward p=0.065, t=2.63 backward p=0.641, t=1.61, post-learning: forward p=0.082, t=2.62, backward p=0.566, t=1.731). Additionally, as time lags are dependent on each other, we also calculated a one-sample-cluster-permutation test

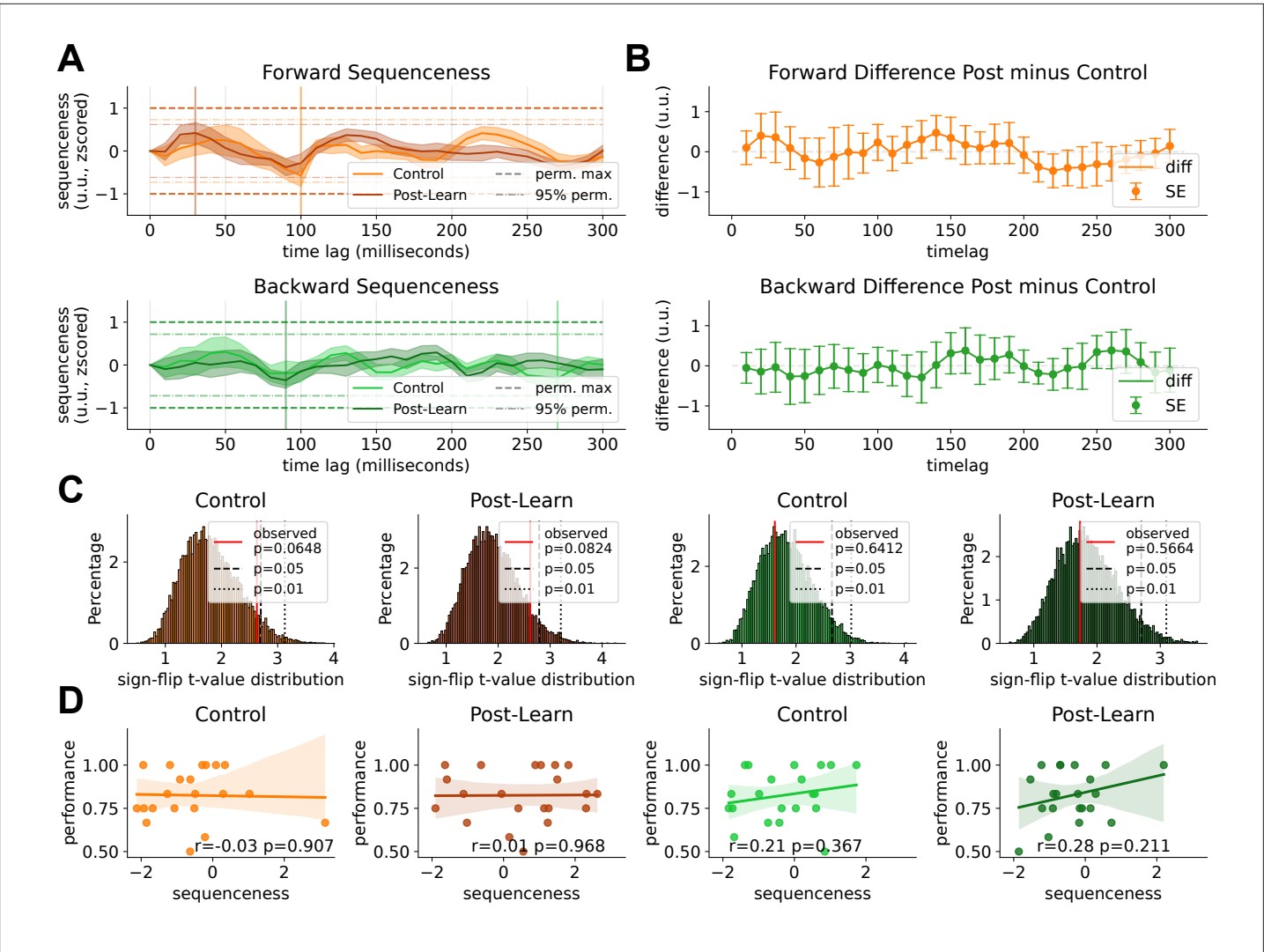

**Figure 3.** Sequenceness result of the resting state sessions. (**A**) Forward and backward sequenceness values on the y-axis at different time lags (x-axis) for control resting state (light colored) and post-learning resting state (dark colored). Two significance thresholds are indicated, the more lenient 95% quantile of the maximum sequenceness of all permutations (lightly dotted line) and the maximum mean peak sequenceness across all permutations of all subjects (bold dotted line). None of the thresholds were surpassed. The peak mean sequenceness has been marked by a vertical line, respectively. (**B**) Difference between pre-learning and post-learning resting states across time lags. Using FDR-corrected paired t-tests for each time lag revealed no significant difference between conditions (without correction, two sequenceness differences were significant: forward 140 ms, p=0.039 and 220 ms, p=0.035). An additionally performed cluster-permutation test found a non-significant cluster at 130–140 ms forward direction (p=0.43). Bands indicate two standard errors of the mean of the difference scores. (**C**) T-value distribution of a 10,000 iterations-sign-flip permutation test. Per permutation, a random subset of subjects' sequenceness values was flipped and the maximum t-value across lags used to construct the empirical distribution. The true maximum t-value indicated in red (nonsignificant in all cases) (**D**) Individual participant's retrieval memory performance relative to their sequenceness values at the peak mean sequenceness (indicated in A). Regression lines and 95% confidence intervals are fitted. For correlation with performance deltas between post and pre-resting state retrievals, see *Figure 3—figure supplement 3*. NB, however, as our simulation shows, correlations of sequenceness with behavioral markers are likely to be underpowered and occur only with very high replay rates or much higher sample size. See our simulation discussion for a more detailed explanation on how correlations may be inherently biased, where fluctuations in baseline sequenceness overshadow individual scaling with behavioral markers. For individual sequenceness curves, see *Figure 3—figure supplement 1*. For within-session forward minus backward sequenceness, see *Figure 3—figure supplement 4*. For sequenceness curves without alpha correction, see *Figure 3—figure supplement 2*.

The online version of this article includes the following figure supplement(s) for figure 3:

**Figure supplement 1.** Individual sequenceness curves for all participants in the control resting state (left) and the post-learning resting state (right), for forward (upper), and backward sequenceness (bottom).

**Figure supplement 2.** Sequenceness curves when not correcting for alpha oscillations.

*Figure 3 continued on next page*

*Figure 3 continued*

**Figure supplement 3.** Individual participant's retrieval performance delta of the post-rest retrieval to the pre-rest retrieval relative to their sequenceness values at the peak mean sequenceness (indicated in upper panels).

**Figure supplement 4.** Forward minus backward sequenceness within each resting state session.

of difference values with 10,000 permutations that considers the temporal correlation of time lag bins. This revealed a single forward non-significant cluster at 130–140 ms (p=0.039) ms (p=0.44). For a visualization of the reported differences, see *Figure 3B*. Finally, we report forward minus backward sequenceness as well as our motivation for an across-session post-pre comparison, instead of within-session forward-backward in *Figure 3—figure supplement 4*.

Additionally, we ascertained whether there was an interaction between memory retrieval performance and individual sequenceness strength by calculating the time lag of peak sequenceness across all participants (control forward = 100ms, control backward: 10 ms, post-learning forward: 280 ms, backward: 240 ms) and correlating individual sequenceness values with participants' performance. There was no significant correlation between peak sequenceness scores and memory performance at post-learning resting state (forward *r*=0.0, p=0.99, backward: *r*=0.02, p=0.935, see *Figure 3D*), nor with the performance difference between post-rest retrieval minus the performance of the last block before rest (see *Figure 3—figure supplement 3*).

Lastly, similar to *Liu et al., 2019*, we split the 8-min resting state into discrete time bins and assessed replay strength for each segment separately (*Figure 4*). In none of these bins was a maximum

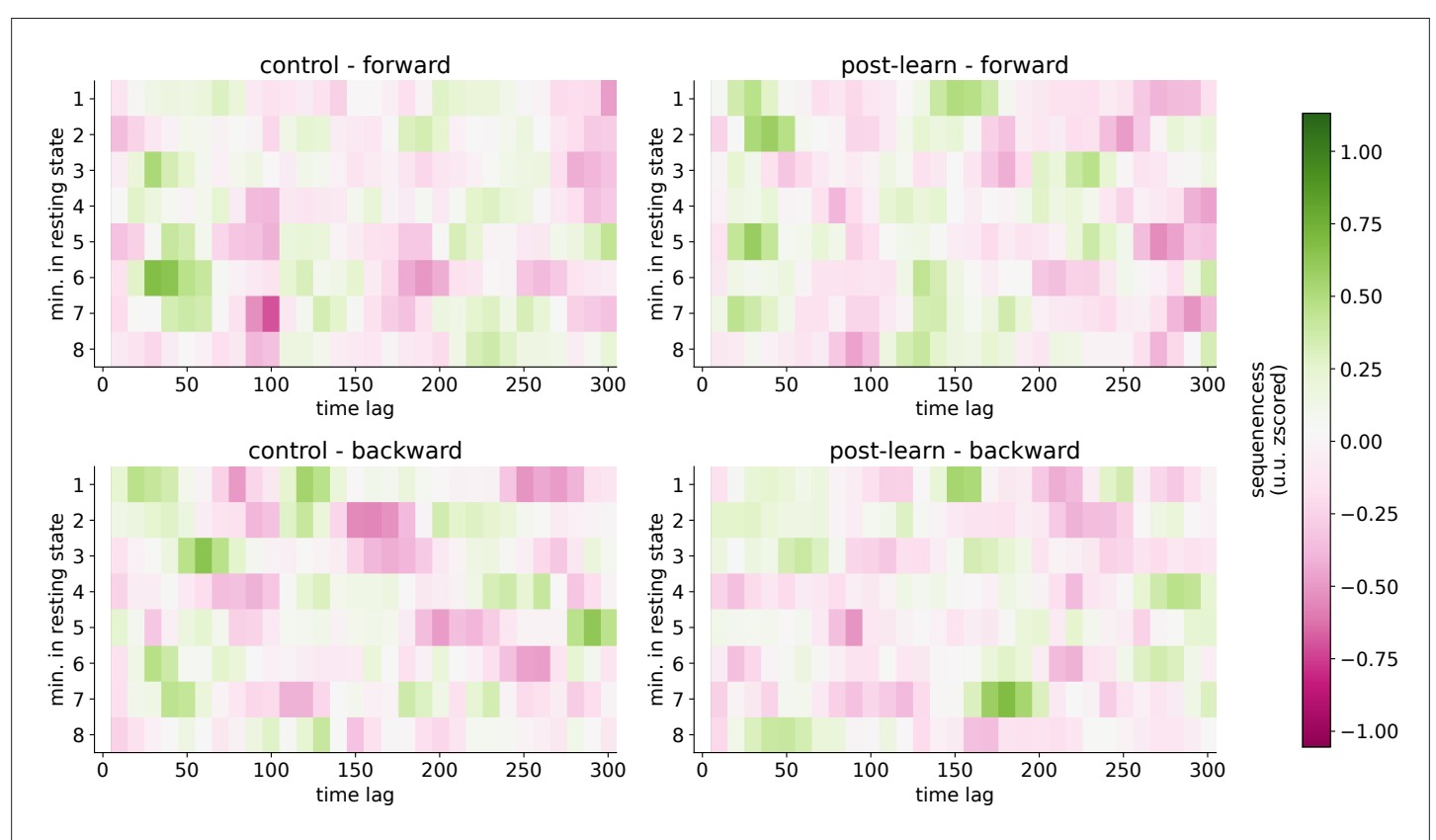

**Figure 4.** Sequenceness map of different segments. Each of the 8 min of resting state is on the y-axis, and different time lags are on the x-axis. For each minute, we took either the first or second half of 30 s for the final calculation and the other 30 s segment for the exploration data set. Sequenceness crossed the 95% permutation threshold in the control resting state in minute 4 (forward 100 ms time lag) alone. However, the sequenceness score was negative, rendering its interpretation non-trivial. Note that we did not apply any control for multiple comparisons across the segments. For the sequenceness curve per segment, see *Figure 4—figure supplement 1*.

The online version of this article includes the following figure supplement(s) for figure 4:

**Figure supplement 1.** Individual sequenceness curves for different segments.

permutation threshold exceeded. A more lenient 95% permutation threshold was exceeded in one segment of the control resting state for minute 4 (100 ms backward sequenceness). However, here the sequenceness value was negative, suggesting, if anything, suppression of replay. Individual sequenceness curves for all segments can be found in *Figure 3—figure supplement 1*.

## Interim discussion

Overall, in resting state data, recorded directly after a learning task block, we did not find evidence for sequenceness using TDLM. This involved implementing previously described protocols in terms of localizer recording, pre-processing and decoder settings, albeit within a different cognitive protocol compared to those previously implemented within resting states (*Liu et al., 2019*; *Nour et al., 2021*; *Nour et al., 2023*; *Wimmer et al., 2020*). Even though we found a single time lag with putative replay during one of the 30-s segments, the evidence here was weak and its validity is questionable. Firstly, multiple comparison correction would have removed the effect. Second, interpretation of negative sequenceness values is unclear. As forward and backward sequenceness are estimated separately, in our interpretation, negative forward sequenceness suggests an expected sequenceness is replayed *less* than expected, which could be interpreted as replay suppression. One possible way this could occur is via a repetition suppression process (*Schwartenbeck et al., 2023*), if one assumes that activation of one item occurs simultaneously with the next in line and subsequently that pattern of neuronal activity is suppressed. However, this appears to us, at best, a convoluted argument. Additionally, recent reports suggest replay avoids self-repetition in addition to avoidance of replay for just experienced episodes (*Mallory et al., 2025*), which would show as negative sequenceness. However, this effect has so far been shown to occur only at a time scale of seconds and not minutes.

Previous reports involving TDLM analysis of resting state data have involved paradigms with a dominant inference component (*Liu et al., 2019*; *Liu et al., 2021b*; *Nour et al., 2021*). Notably, in our study, we used a declarative graph-based memory paradigm that lacked this inferential component. We note previous research in humans has shown reactivation and replay during rest in the context of non-inferential paradigms using fMRI (*Ogawa et al., 2024*; *Schapiro et al., 2018*; *Tambini and Davachi, 2013*), and especially in the context of declarative learning (using fMRI and EEG: *Deuker et al., 2013*; using intracortical electrodes: *Eichenlaub et al., 2020*; using fMRI: *Schlichting and Preston, 2014*; using intracranial EEG: *Zhang et al., 2018*). On the other hand, it has been reported that replay is more likely to be expressed for weakly encoded memories (*Denis et al., 2020*; *Schapiro et al., 2018*) and hence it might be absent if encoding strength is too strong. While we employed a learning criterion of 80% for this reason, as is usual in the broader context of sleep and memory research, we cannot be certain if this represents a level of encoding strength that lessens a requirement for task-related replay in our study. Anecdotal evidence for replay occurrence might come from many participants reporting involuntarily seeing segments of the sequence flashing in front of their eyes during the resting state, although we caution that there is mixed evidence for a link between anecdotal reports of conscious experience and the occurrence of replay (*Komsky, 2015*; *Wittkuhn et al., 2024*).

Additionally, given that the stimuli were presented in combined triplets, participants may have formed a singular representation of associated items and subsequently replayed these (e.g. AB→C), instead of replaying item-by-item transitions (A→B→C). Under such a scenario, a classifier trained on individual items would not detect these newly formed representations, particularly if they diverge strongly from single-item patterns. In our previous study, where we addressed retrieval (*Kern et al., 2024*), we found that states were to varying extents co-reactivated, yet classifiers trained on single items retained sensitivity to detect these combined reactivation events. Consistently, prior work suggests that unified representations retain overlap with their constituent item representations (*Dennis et al., 2024*; *Liang et al., 2020*). However, there is also evidence that different brain regions are involved if unitization occurs (*Staresina and Davachi, 2010*), potentially confusing classifiers. Therefore, we cannot exclude that rest-related consolidation replays engendered unitized representations that were insufficiently captured by our single-item classifiers.

As we did not know whether the absence of replay was inherent to our paradigm, in a second study, we explored potential explanations for this null finding. Specifically, we conducted a hybrid simulation study that combined localizer data and data from a control resting state. In the context

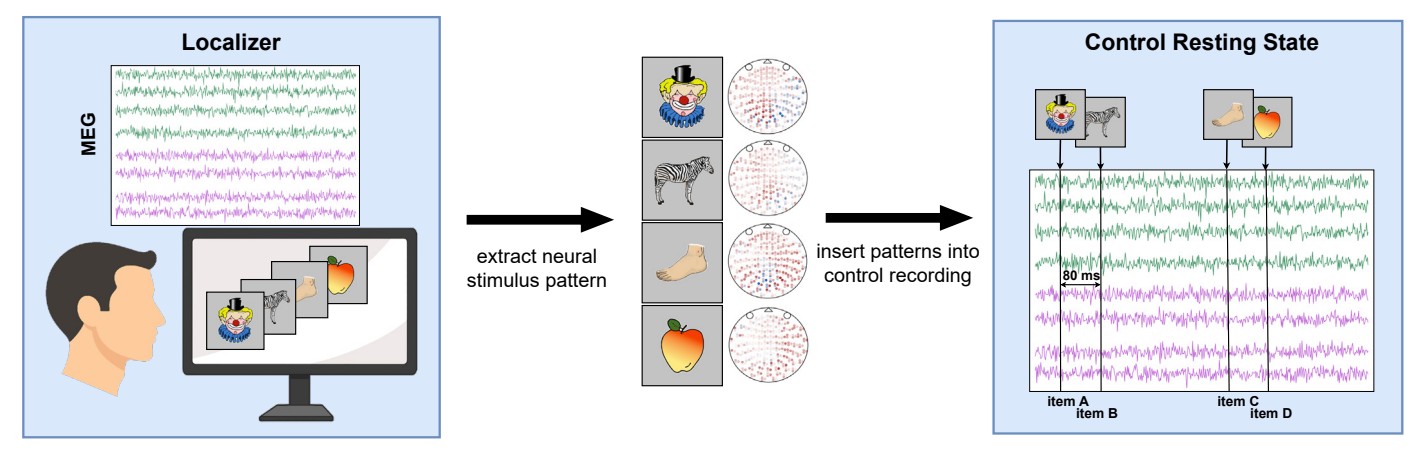

**Figure 5.** Schematic of procedure for inserting simulated replay into the control resting state. First, neural patterns are extracted, for each stimulus, from the peak of decodability during the localizer task. Participant-specific patterns are normalized by subtracting the average sensor values of seeing any other stimulus from the sensor values of seeing only the stimulus of interest. Next, these subtle patterns are inserted into the control resting state at a fixed interval, following a one-step transition pattern according to the task sequence. A refractory period is retained between each replay event to prevent overlaps. For details, see Materials and methods section. For a visualization of the mean sensor patterns that have been inserted, see *Figure 5—figure supplement 1*.

The online version of this article includes the following figure supplement(s) for figure 5:

**Figure supplement 1.** Visualization of the sensor patterns that were inserted into the resting state per class, averaged across all participants.

of our paradigm, this simulation allows quantification of the strength of replay necessary for it to be detectable using the TDLM methodology.

## Results study II: simulation

In a hybrid simulation study, we enriched the control resting state data of participants with simulated replay events. That means we inserted sequences of neural patterns extracted from the localizer (taken from the time point of peak decoding accuracy) into the control resting state with a state-to-state lag of 80 ms. We chose this timepoint (80 ms state-to-state lag) as its sequenceness value was close to zero in the baseline condition as well as it being distant to notable alpha rhythms of participants (which varied between ~9 and 11 Hz). Additionally, we subtracted the mean sequenceness value of the baseline at 80 ms lag such that any simulation effects would, on average, start at zero sequenceness.

We varied the density of replay, traditionally denoted as 'replay events per minute' ($min^{-1}$) and looked at the resulting sequenceness strengths. Additionally, we scaled the replay density inversely with the participants' retrieval memory performance to simulate a putative increase of replay for less stable memories. For example, a participant with the lowest memory performance (50%) would have a factor 1.0 applied to their replay amount, while a participant with the highest memory performance (100%) would show reduced replay and have a scaling factor of 0.5 applied to the number of inserted replays. We used different scaling factors to explore how they contribute to relationships between measures of sequenceness and memory performance. While we acknowledge the relation between replay and memory performance is probably more complex (*Kern et al., 2024*; *Schapiro et al., 2018*; *Wimmer et al., 2020*), we applied this simplified scaling to assess whether TDLM can detect behavioral correlations with sequenceness strength, as has been shown previously (*Kern et al., 2024*; *Liu et al., 2019*; *Nour et al., 2021*; *Wimmer et al., 2020*; *Yu et al., 2023*). Our results indicate that replay and correlations with behavior were only detectable at very high - putatively biologically implausible - densities (*Figure 5*).

## Sequenceness is only detectable at very high replay density

Our simulation analysis found that a replay density of 130 $min^{-1}$ and a density of ~80 $min^{-1}$ were necessary in order to reach the maximum and 95% quantile permutation threshold, respectively (see

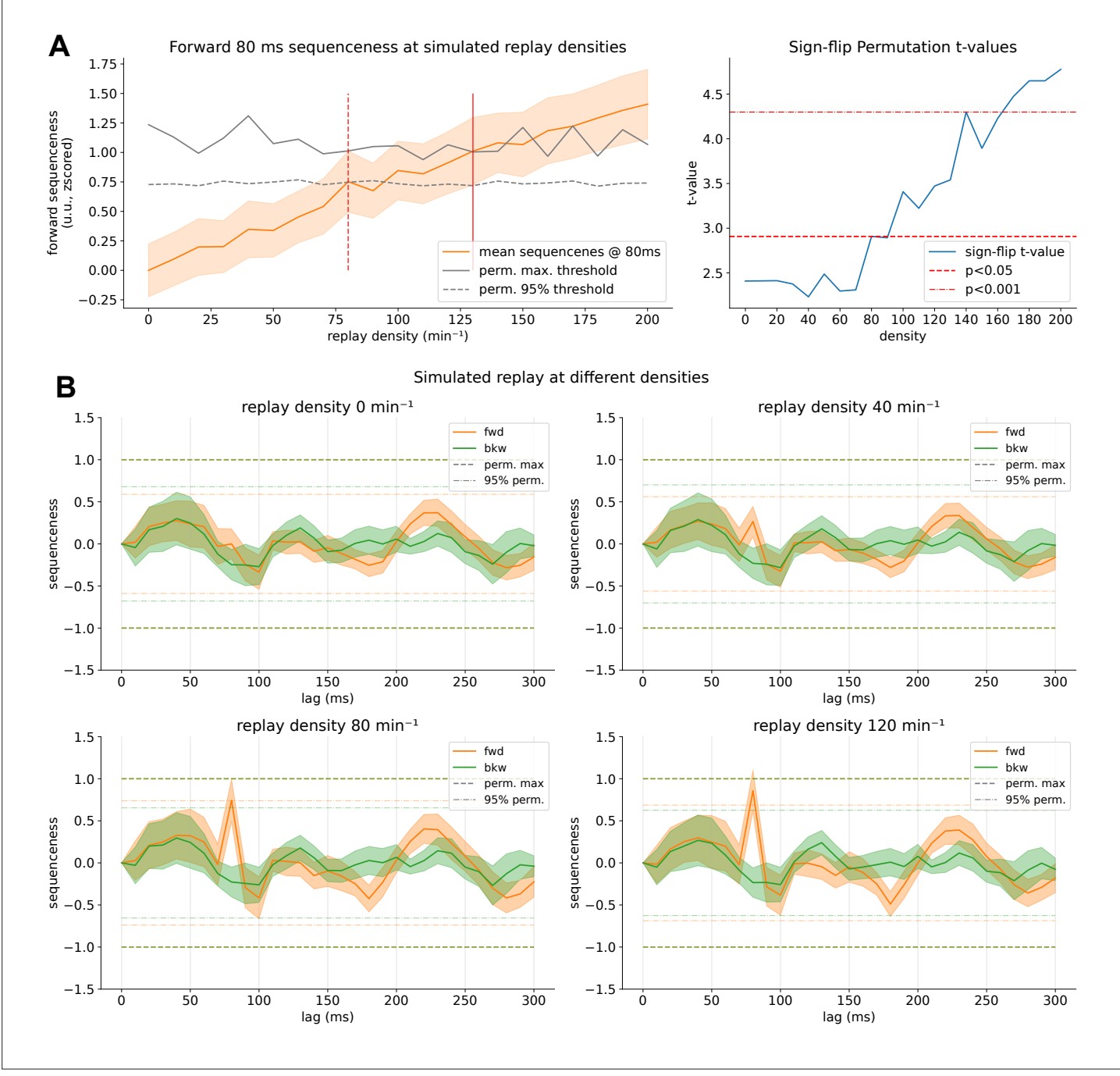

**Figure 6.** Sequenceness results of the hybrid simulation. (**A**) Simulation results for inserting replay at different densities. On the y-axis, sequenceness at 80 ms state-to-state time lag, at different replay densities on the x-axis with standard error as shaded orange. The 95% confidence interval significance threshold is reached at a density of ~80 min⁻¹, while the maximum permutation threshold is reached only around 130 min⁻¹. (**B**) Observed t-values and significance thresholds of a sign-flip permutation test, in which the maximum t-value across time lags for sign-flip permutated sequenceness curves are used to construct a non-parametric null distribution (see Materials and methods for details). Significance (p<0.05) is reached similarly at around 80 min⁻¹. (**C**) Sequenceness across time lags for replay at baseline (0 min⁻¹) and simulated for a density of 40, 80, and 120 min⁻¹ with a simulated time lag of 80 ms. In this simulation, we assumed a temporally uniform distribution of replay throughout the resting state, as TDLM is relatively invariant to temporal distribution (see Appendix). For a power analysis with varying bootstrapped participant sample sizes, see *Figure 6—figure supplement 1*.

The online version of this article includes the following figure supplement(s) for figure 6:

**Figure supplement 1.** Power analysis of sequenceness significance for bootstrapped sample sizes.

*Figure 6A*). We also performed the above-mentioned sign-flip permutation test on the simulated data, reaching comparable results, with significance reached at around 80 min$^{-1}$ (see *Figure 6B*). *Figure 6C* shows sequenceness curves for selected densities at 0 min$^{-1}$ (baseline), 40 min$^{-1}$, 80 min$^{-1}$, and 120 min$^{-1}$. At 40 min$^{-1}$ only a small increase of sequenceness above baseline is detectable. Note the significant drop of sequenceness at 100 ms, which is due to the alpha control introduced at 10 Hz by inserting repeating patterns of the decoded items (*Liu et al., 2021a*). The alpha correction likely over-corrected and introduced negative sequenceness at 10 Hz seen in some participants. Therefore, we include the analysis without alpha-correction in *Figure 3—figure supplement 2*. NB the strong negative sequenceness at 100 ms time lag was not present in our exploratory data set and only appeared in data held back for the final calculation (as preregistered, we used 50% of the resting state data for exploration and parameter search and report here on the unseen 50% after parameters were chosen, see Materials and methods section for details).

## Correlation with behavior only present at high replay speeds

The correlation of behavior with sequenceness at 80 ms time lag was non-significant, even at the maximum replay densities of 200 min$^{-1}$ (see *Figure 7A*). Here we assumed a realistic scenario that linearly downscales replay density with increasing performance from 100% down to 50% for increased memory performance. To amplify the correlation effect, in a secondary analysis, we normalized the performance scaling between 100% and 0% (i.e. 100% performance would lead to 0% of replay inserted). In this maximum-scaling scenario, the correlation of sequenceness with performance was markedly boosted, yet did not reach significance even at a density of 200 events min$^{-1}$. When further investigating this lack of influence of our scaling, we found that even in the baseline condition without simulated replay (0 min$^{-1}$), the correlation with performance seems to oscillate, depending on which time lag it is computed on (*Figure 7B*), indicating fluctuations of sequenceness between participants that are non-random. If we chose a different time lag for our simulation, added effects of baseline plus simulation would have probably reached significance more easily (e.g. at the 240 ms time lag). This baseline variation becomes apparent when comparing sequenceness of participants at baseline with sequenceness values boosted via simulated replay at 80 ms time lag. Even in the baseline condition (0 events min$^{-1}$, *Figure 7D*), some participants express higher sequenceness values than others with simulated replay of 160 min$^{-1}$. *This indicates that the effect* that we wanted to introduce via scaling the inserted events with memory performance is overshadowed by some form of bias present at baseline, that induces marked variations in sequenceness without any replay present, even after a recommended GLM-based alpha control (*Liu et al., 2021a*).

The resulting correlation is of moderate size even at higher densities (Pearson's r of around 0.3), indicating our paradigm was likely underpowered for this specific analysis. Therefore, we performed a bootstrapped power analysis by oversampling existing participants, with 10,000 repetitions per sample size. We found that more than 160 participants would have been necessary to detect a correlation between performance and sequenceness in the simulated density of 100 min$^{-1}$ condition (*Figure 7C*) with 80% power where the above-mentioned maximum scaling applied. Nevertheless, it is worth bearing in mind that this might also lead to increased power to detect nuisance effects in the baseline condition, if the effect is added to a pre-existing correlation in the baseline condition.

## Previous synthetic simulation overestimates TDLM sensitivity

Next, we compared our hybrid simulation to a purely synthetic TDLM simulation, closely following the demonstration code in *Liu et al., 2021a*. We show that in the absence of endogenous oscillatory patterns, and using synthetic patterns, significance thresholds are easily surpassed at around 10–20 min$^{-1}$ (*Figure 8A*). In this setting, synthetic resting state data is generated autoregressively by sampling values from a multivariate normal distribution given a sensor covariance matrix. The class-specific stimulus patterns are created by first drawing a random sensor pattern common to all classes (akin to an ERP) and adding a random class-specific pattern for each class, plus noise for each trial of that class. Null data is created by drawing from a normal distribution. For the full procedure, see Materials and methods Study II. Note, for consistency, we used the same insertion logic and code as for the hybrid simulation, which deviates slightly from the MATLAB implementation and simulates with an unrealistic 2000 min$^{-1}$ (see *Figure 8—figure supplement 1*).

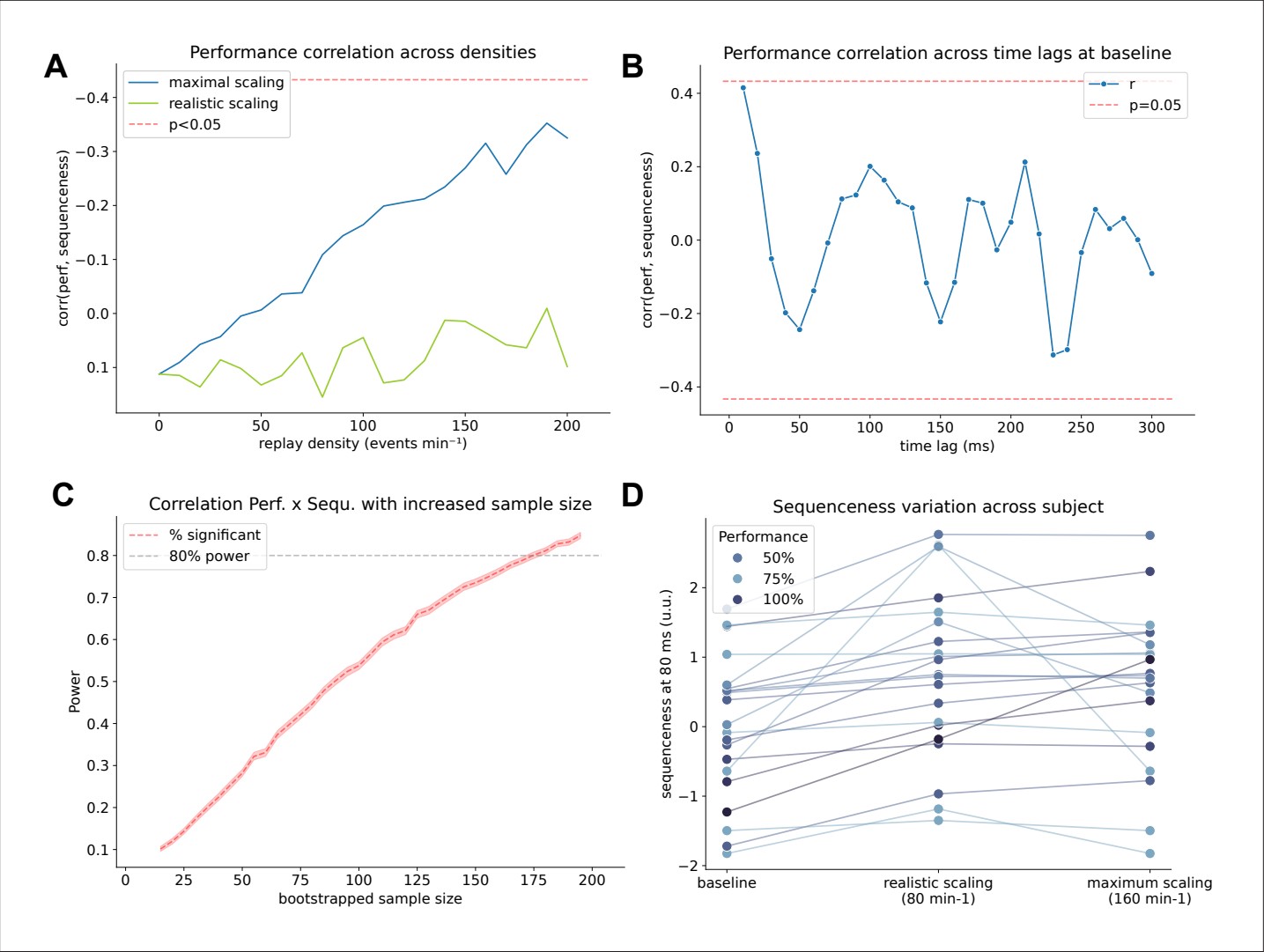

**Figure 7.** Correlation with behavior is underpowered and strongly affected by systematic baseline fluctuations. (**A**) Relationship (Pearson's r) between memory performance (behavior) and sequenceness at 80ms time lag on the y axis, with replay density on the x-axis for two scaling rules: Realistic scaling: Replay density was inversely scaled with participant's performance to between 50 and 100% (i.e. the worst performer had 100% while the best performer only had 50% of replay density). Maximal scaling: Replay density was scaled in from 0% and 100% (i.e. the best performer had no replay inserted). Significance of the correlation was not reached even at the maximum density. However, note that the baseline correlation started at $r=0.1$, and simulated effect without this offset would have reached significance at a density of around 200 min$^{-1}$. (**B**) Correlation of sequenceness computed across different time lags in the control condition (i.e. without inserted replay). Even when no replay is present, the correlation fluctuates significantly depending on which time lag it has been computed, possibly due to baseline variations that fluctuate differentially between participants. (**C**) Bootstrap power analysis showing how many participants would be necessary to detect a significant correlation between replay and performance in the maximally scaled scenario with a replay density of 130 min$^{-1}$. 80% power to detect a correlation is reached around a sample size of around 160. (**D**) Sequenceness at 80 ms time lag in the control condition (left), in the realistic scenario with replay scaling from 50% to 100% (middle) and the maximal scenario with scaling from 0% and 100% (right). Participant's performance is indicated by increasing saturation of blue. While all participants show an increase in sequenceness with increased scaling, some participants have a higher sequenceness value and an ordering according to performance is barely visible in the maximum scaling case.

The maximum permutation threshold is reached already at a density of 13 min$^{-1}$ (*Figure 8A*), similarly to the original MATLAB code (see *Figure 8—figure supplement 1*). The 95% of permutation sequenceness maxima is reached at 9 min$^{-1}$. Similarly, the sign-flip permutation threshold is reached at 6 min$^{-1}$, showing again an increased sensitivity of this new permutation approach. Individual sequenceness plots (*Figure 8B*) show that the purely synthetic setup produces robust sequenceness effects at substantially lower replay densities than evident in our hybrid simulation, surpassing thresholds

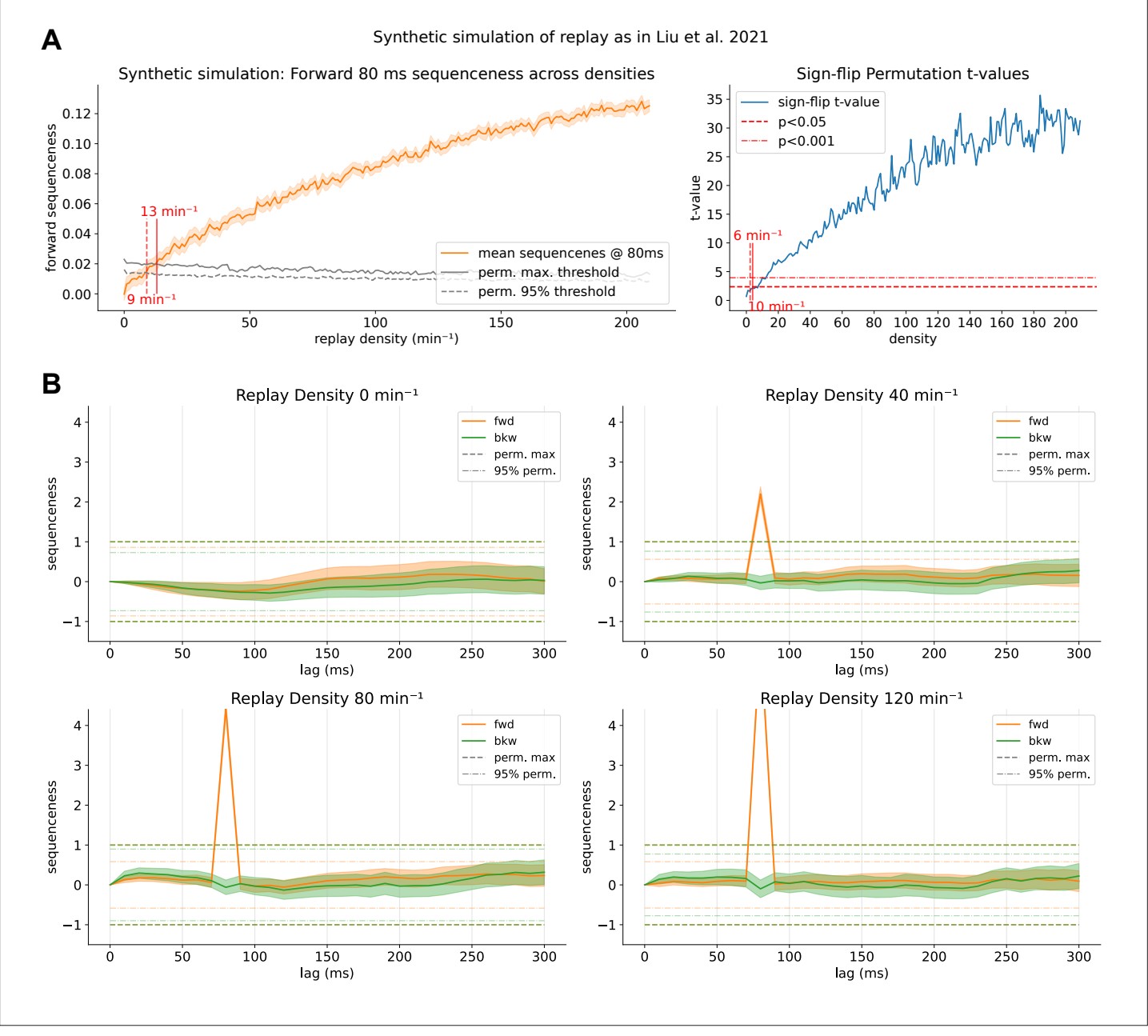

**Figure 8.** Results of the fully synthetic simulation, akin to the one presented in *Liu et al., 2021a*. (**A**) Mean sequenceness at the simulated time lag of 80 ms. Thresholds are reached already at a much lower density of a rate of ~13 events/min, in line with the original MATLAB code (see *Figure 8—figure supplement 1*). (**B**) Four exemplar plots for sequenceness at a density of 0 (baseline), 40, 80, and 120 events/ min. Oscillatory behavior of the sequenceness curves is very minimal in the baseline condition. Maximum permutation thresholds are reached easily in the 40 events/min case and surpassed several times with higher densities. However, exactly the lack of these spurious confounders and a too strong pattern discriminability make it possible for TDLM to work well in a synthetic dataset. See *Figure 8—figure supplement 1* to compare the result to the original MATLAB simulation code.

The online version of this article includes the following figure supplement(s) for figure 8:

**Figure supplement 1.** Rerun of original MATLAB simulation code.

manifold even for lower sequenceness. However, importantly, when examining the baseline condition, a notable lack of oscillatory behavior is observed compared to empirical data (*Figure 8B* upper left). To quantify why sequenceness emerged more readily in the synthetic case, we compared decoder output distributions at timepoints where a true pattern is present. Therefore, we extracted classifier

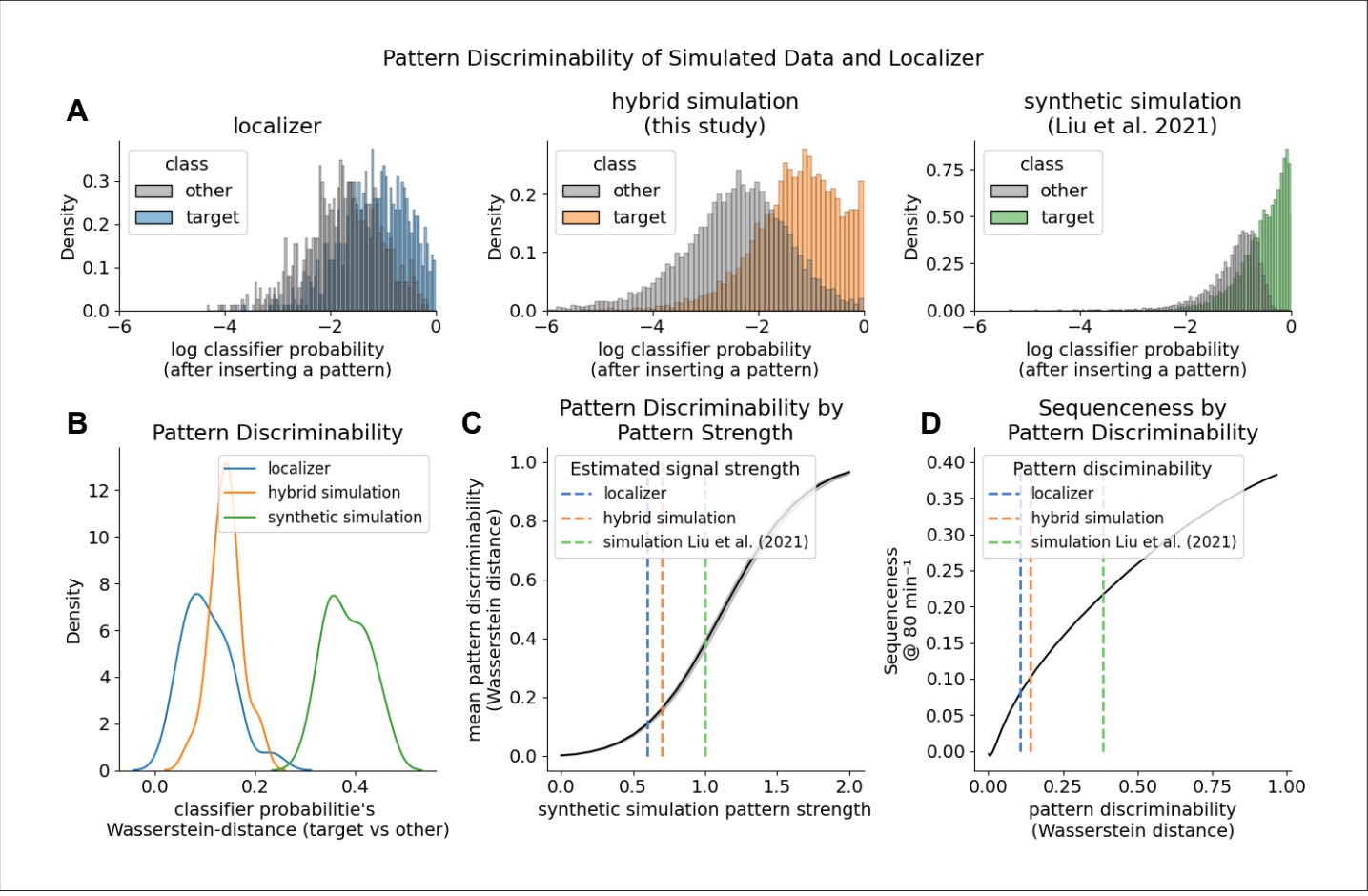

**Figure 9.** Pattern discriminability between the target class and other classes impacts sequenceness sensitivity. (**A**) Histograms of log-scaled classifier probability estimates during the peak decodability of localizer (left, cross-validated), for reactivation events in the hybrid simulation (middle) and synthetic simulation (right). (**B**) KDE plot of the Wasserstein distance of participants' classifier probabilities. The hybrid simulation likely over-estimates how well the decoder finds a reactivation event, while the synthetic simulation largely over-estimates. (**C**) Pattern discriminability as a function of the inserted pattern strength (akin signal-to-noise ratio). We scaled the magnitude of the inserted pattern in the synthetic simulation. Dotted lines indicate estimated pattern strength of the empirical data and hybrid simulation as given by their mean Wasserstein distance (**D**) TDLM sequenceness increases with better pattern discriminability. However, it is questionable if the pattern discriminability during a resting state reactivation event is higher than during the localizer.

probabilities given to the true class and all other classes. *Figure 9A* shows histograms of log-scaled classifier probability estimates for the peak-decoding time point in the localizer (left, cross-validated) and during a simulated reactivation event in both the hybrid simulation (middle) and synthetic simulation. Classifier probabilities of the target class in the synthetic simulation were substantially more extreme, indicating stronger separation between target and non-target patterns than in the empirical localizer. Next, we quantified the difference between the target and other probability estimate distributions using the Wasserstein distance (*Villani, 2009*) per participant (*Figure 9B*), as a higher distance should result in better discriminability. Pattern discriminability was lowest in the localizer trial ($W_1=0.11$), intermediate in the hybrid simulation ($W_1=0.14$; consistent with the fact that inserted patterns were derived from empirical data) and markedly increased for the synthetic simulation ($W_1=0.38$). Notably, both simulations overestimate pattern separability, as we hypothesize that the pattern discriminability during a real reactivation event is less pronounced than in the localizer during the strong signal induced by visual image processing.

Next, we tested what would happen if we increased the strength of the inserted pattern. Therefore, we scaled the synthetic pattern by a factor of 0–2 before insertion and measured the Wasserstein distance of the resulting decoder output. Unsurprisingly, the stronger the magnitude of the inserted pattern, the better the discriminability between the target and other class (*Figure 9C*). This allowed

us a rough estimate of the pattern strength during the localizer and hybrid simulation by mapping their Wasserstein distances to the strength in the synthetic scaled simulation. Next, we compared how increased pattern discriminability would affect sequenceness (*Figure 9D*). We simulated synthetic replay as above at a density of 80 min⁻¹. As expected, the higher the discriminability between the target class and other classes, the higher the sequenceness. Thus, greater discriminability will lead to more extreme classifier outputs, increasing the beta weights of the first-level GLM of TDLM.

Overall, our results indicate that the purely synthetic simulation produces substantially higher pattern discriminability than observed in the empirical localizer and the hybrid simulation. This enhanced discriminability is accompanied by a strong increase in sequenceness magnitude, suggesting that previous synthetic models may provide an over-optimistic estimate of TDLM's sensitivity. Additionally, by omitting endogenous oscillatory confounders, synthetic simulations fail to account for spurious baseline sequenceness that confound empirical detection.

## Discussion

Using TDLM, we investigated whether sequential replay of learned graph-structured associations could be detected during a post-learning resting state. Contrary to expectation based on previous findings, we did not find evidence for replay of well-learned sequences during a post-learning rest period. This was despite using a similar methodology to that used in prior replay investigations of resting state, albeit these involved different paradigms (*Eldar et al., 2020*; *Huang et al., 2024*; *Liu et al., 2019*; *Nour et al., 2021*; *Yu et al., 2023*). However, due to differences from previously used paradigms, we cannot conclude that lack of evidence for replay reflects the sensitivity of our detection method (TDLM) or reflects that our task was not eliciting sufficiently strong resting state replay to enable replay detection using TDLM.

Based on the above, we designed a simulation study to explore the strength of replay being necessary for successful detection. This simulation study also speaks to the potential sensitivity of TDLM as an explanation of our null findings. By inserting synthetic replay events into pre-task resting state data, we show that replay is only detectable under an assumption of very high replay rates—with more than 80 replay events per minute being necessary to reach significance thresholds. Here, one 'replay event' is defined as a train of sequential reactivations with a certain state-to-state time lag, that is 80 ms, meaning more than one replay event per second on average throughout the entire resting state. Furthermore, we find that already at baseline (i.e. without any replay) sequenceness values are biased by, yet unknown, confounders that introduce absolute offsets in baseline sequenceness values between participants. We contrast our novel hybrid approach to a purely synthetic simulation and show that previously proposed approaches are likely to overestimate pattern discriminability and endogenous neural patterns as oscillations, overestimating TDLM's sensitivity.

Our hybrid simulation indicates that when an increase in sequenceness by actual replay is introduced, the baseline offset between participants is larger than the increase at the group level. Therefore, correlating behavioral measures with individual sequenceness values should be regarded with caution unless this baseline offset is controlled for. Our findings highlight a potential need for further methodological refinements in both experimental paradigms and analysis techniques to reliably detect human replay, using non-invasive measures, during extended offline periods (such as resting state or sleep). Finally, we make recommendations for future studies and suggest ways to enhance the reliability of replay detection methods.

### High replay density necessary to detect sequenceness

In our study, more than one replay event per second (80 min⁻¹) would have been necessary to detect replay at a group level. Sharp-wave-ripples (SWR) are traditionally considered substrates for representational reactivation and hence strong replay expression (*Eschenko et al., 2008*; *Joo and Frank, 2018*; *Roumis and Frank, 2015*), with each SWR containing several sequential item reactivations. While every SWR might not contain task-relevant activity (*Schreiner et al., 2024*), their occurrence rate is considered a reasonable proxy for estimating in vivo replay density. Previous research in rodents has reported that ripple rate significantly varies between tasks and states, being most prominent in offline states such as NREM and post-task rest with a replay density of around 20–25 events per min⁻¹ (*Eschenko et al., 2008*; *Ponomarenko et al., 2003*; *Wiegand et al., 2016*). In humans, evidence

from intracranial recordings is sparse, but studies estimate a ripple density of between 1 and 15 min$^{-1}$ during NREM sleep (*Jiang et al., 2020*; *Ngo et al., 2020*) and ~1.5–4 min$^{-1}$ during resting state (*Chen et al., 2021*). However, it should be noted that reported ripple rates depend on an applied detection criteria and so might underestimate their actual true occurrence (*Liu et al., 2022a*). While in vitro studies have reported that CA1/CA3 neurons can maintain a (experimentally induced) density of up to 50–200 min$^{-1}$ (*Buzsáki, 2015*; *Jiang et al., 2018*; *Kanak et al., 2013*), to our knowledge, such densities have not been reported in vivo.

As replay was detectable at around 80 min$^{-1}$ (~1.33 replay events per second), with more strict thresholds only being reached at around 130 min$^{-1}$ (~2.5 replay events per second). Based on previous research, it might be considered implausible that these densities would be maintained throughout an 8-min resting state. Nevertheless, as it is known that SWRs occur in bursts (*Buzsáki, 2015*), and it is conceivable that higher ripple rates are maintained during shorter time windows, as for example during short intervals of recall, where they can reach more than 40 min$^{-1}$ (*Norman et al., 2019*).

Interestingly, *Higgins et al., 2021* report that replay events detected by TDLM were not evenly distributed throughout the resting state and, even though not explicitly reported, varied between 9 and 40 min$^{-1}$ for different segments of an exemplar participant. When replay occurs in bursts, certain time windows might reach necessary density thresholds, while averaging over larger windows might be inconclusive. However, such high-density windows would need to be selected on a principled basis to avoid inflating false positive rates. Nevertheless, in our post-hoc analogue sub-analysis of 30 s individual segments of the resting state, we found no clear evidence for replay. Currently, in the context of MEG replay studies, it is not reported what replay rates (events min$^{-1}$) are assumed. Therefore, future studies might consider providing estimates for replay rates under which reported findings are plausible, analogous to SWR rates reported in animal studies, for example via simulations.

However, while we use indices of SWRs as a proxy for replay density estimation, the relationship between hippocampal replay and replay detected by TDLM remains uncertain. Current decoding approaches measure replay-like phenomena over cortical sites, with previous reports highlighting a spectral power increase in hippocampal areas coinciding with TDLM detected replay episodes (e.g. *Liu et al., 2019*; *Wimmer et al., 2023*). Nevertheless, it is conceivable that cortical replay detected by TDLM might occur independent of hippocampal replay and SWRs, as well as being generated by different mechanisms. Furthermore, some TDLM studies find a replay state-to-state time lag of above 100 ms (*Huang and Luo, 2023*; *Liu et al., 2021b*; *Wimmer et al., 2020*; *Wimmer et al., 2023*), which is much slower than previously reported place cell replay. One obvious conclusion is that we know too little about the characteristics of replay as detected by TDLM, specifically with respect to its relationship to SWRs. Future studies might employ simultaneous intracranial and cortical surface recordings to throw light on the precise relationship between hippocampal replay and replay as detected by TDLM. Similarly, in our simulation, for simplicity and to maintain consistency with previous stimulations, we restricted replay events to those spanning two reactivation events. While the characteristics of replay as measured by TDLM are unknown, it is conceivable that several event steps can be replayed within one replay event. We show that the vanilla version of TDLM is fundamentally sensitive to the number of single-step transitions alone and disregards if these are either replayed, chained, or chunked (Appendix and *Appendix 1—figure 2*). Nevertheless, if the number of reactivation events chained within a replay event increases, TDLM sensitivity is increased relative to the replay density and thresholds are reached earlier (see *Appendix 1—figure 3*). See *Appendix 1—figure 4* for a simulation of multi-step replay events and our discussion of the caveats.

## Correlation with behavior is overshadowed by baseline fluctuation of sequenceness

The occurrence of replay, in the absence of ground truth, can to some extent be validated by a correlation between a replay measure and behavioral performance (*Liu et al., 2021b*; *Wimmer et al., 2020*). In our simulation, we could not induce a strong relationship of sequenceness to participants' performance at realistic replay densities. Even in an unrealistic, maximal-scaling scenario, with no replay for the best-performing and 100% replay for the worst-performing participant, correlations were only significant at implausibly high replay densities of around 200 min$^{-1}$. A power analysis revealed that around 160 participants would be necessary to establish a correlation in the maximum scaling case at a density of 80 min$^{-1}$. We find that the magnitude of baseline sequenceness values across participants

was higher than the induced effects and was hence not affected strongly by inserting synthetic replay (see *Figure 7D*). Sample sizes in previous TDLM literature usually range between 20 and 40 participants. A bootstrap power analysis reveals that even at those sample sizes and beyond, power would remain low unless unrealistically high replay rates are assumed (*Figure 6—figure supplement 1*). Additionally, the correlation analysis between sequenceness and behavior would be greatly underpowered for higher sample sizes, even under an assumption of high replay densities (see *Figure 7C*).

## Spurious sequenceness in the control condition

In data without any expected replay, sequenceness should be close to zero, or else be attributed to noise or bias. While random noise should average out with increasingly large number of participants, it remains possible that some systematic bias is shared among participants. For example, this might be the case with alpha oscillations, which occur at roughly the same frequency and tend to increase in strength from pre to post resting state due to participant fatigue. In our study, non-significant fluctuations in sequenceness already occurred in the control condition, a phenomenon not unique to this study (and reported in supplementary material of *Nour et al., 2021*; *Nour et al., 2023* and *Liu et al., 2019*). Thus, it seems as if a yet unknown bias introduces spurious systematic sequenceness effects that, in our simulation, may serve to overshadow actual sequenceness at low densities. While, as previously discussed (*Liu et al., 2021a*; *Liu et al., 2019*), endogenous oscillations such as eyes-closed-alpha can impact TDLM, the spurious (albeit non-significant) sequenceness in our case was not sufficiently controlled by alpha correction to make it disappear completely (*Figure 3—figure supplement 2* for analyses without the 10 Hz correction method). Interestingly, the magnitude of the correlation of performance with sequenceness, as discussed above, fluctuates approximately at a 10 Hz rhythm. This leads us to speculate that alpha oscillations may impact baseline sequenceness in a manner not entirely removed by controlling for alpha using recommended methods (*Liu et al., 2021a*). Furthermore, we speculate that other factors might contribute to spurious sequence detection, for example oscillatory inter-trial coherence during localizer training (*Hoffman et al., 2013*; *Kragel et al., 2020*; *van Diepen and Mazaheri, 2018*), (phase)-coupling of two or more oscillations (*Fell and Axmacher, 2011*; *Watrous et al., 2015*) or phase differences in traveling waves (*Hughes, 1995*; *Muller et al., 2018*). For example, if two classifiers' probabilities oscillate at 10 Hz but at a different phase, a spurious time lag can be found reflecting this phase shift. We speculate that more complex interactions between classifiers oscillating at different phases are also conceivable. Exploring these possibilities is beyond the scope of the current study and should be evaluated systematically in future experiments.

Finally, yet importantly, it is important to consider that endogenous oscillations might play a functional role. Thus, it has been proposed that certain theta phases support encoding and consolidation (*Hasselmo et al., 2002*; *Kerrén et al., 2018*; *Kerrén et al., 2022*), or that oscillations provide a window for reactivation (*Higgins et al., 2021*; *Staresina et al., 2015*). Such a scenario poses a challenge for differentiating between a theta-phase bias, not attributable to memory processing, from actual reactivation present only during a specific phase of an oscillation.

## Probing the conditions under which TDLM works requires realistic simulations

Previous studies performed sequenceness analyses on simulated replay, but replay density has rarely been simulated under realistic conditions. For example, in the sample code of *Liu et al., 2021a*, replay was simulated with a density of 2000 min$^{-1}$. Similarly, *Liu et al., 2021b* a density of 2400 min$^{-1}$ was used. Re-running these simulation scripts, with arguably a more realistic replay density (i.e. 15 min$^{-1}$ for NREM, as found in *Jiang et al., 2020*), sequenceness failed to reach significance, even with purely synthetic data (*Figure 8—figure supplement 1B*). Our recreation of the purely synthetic simulation, closely following *Liu et al., 2021b*, reproduced an impression of high sensitivity, with significance thresholds exceeded at densities on the order of ~10–20 events·min$^{-1}$ (*Figure 8A*). However, in contrast to empirical resting state, the synthetic baseline showed little of the oscillatory and non-stationary structure that drives spurious sequenceness fluctuations in real data (*Figure 8B* upper left). Moreover, synthetic class patterns were substantially more separable than empirical patterns, yielding markedly more extreme classifier probability estimates during reactivation time points than observed in the localizer or in our hybrid simulation (*Figure 9A*). It is especially doubtful whether pattern

discriminability would be higher during a reactivation event than during the localizer at the time-point of peak decodability. Together, these results indicate that purely synthetic simulations are liable to substantially overestimate TDLM sensitivity unless calibrated to match both empirical nuisance structure and empirically observed decoder-pattern separability. A hybrid approach naturally provides both aspects.

TDLM at its core is a parameter-free method. Nevertheless, a large space of decoding modalities exists that can influence an analysis outcome. In this paper, we primarily applied default choices as previously reported (L1 Logistic Regression trained on peak decoding time point as in *Kern et al., 2024*; *Kurth-Nelson et al., 2016*; *Liu et al., 2019*; *Nour et al., 2021*; *Wimmer et al., 2020*; *Wimmer et al., 2023*). Nevertheless, we note many different parameter combinations have been applied in other studies and their combined effect remains unclear (*Eldar et al., 2020*; *Huang and Luo, 2023*; *Wimmer et al., 2020*; *Wise et al., 2021*). For example, here we observed that mean baseline probabilities produced by the classifier vary per participant (*Figure 11—figure supplement 2*). While probabilities increased more or less linearly across all participants, we opted to normalize these by dividing the mean probability per class per participant. Similarly, in an exploration dataset, we observed that sequenceness effects were driven by a few participants who produced high magnitudes of sequenceness scores. To mitigate these effects, we z-scored sequenceness for each trial, as reported previously (*Wimmer et al., 2020*; *Yu et al., 2023*). However, this approach also has its drawbacks, as it could potentially deflate true events in participants with good decodability and so inflate false positives, and vice versa.

Furthermore, many different classifier regimes have been used in the context of TDLM (e.g. Lasso Logistic Regression vs SVM, sensor-based decoding vs. temporal embedding with PCA, normalization of sequenceness scores via z-scoring vs no normalization, using classifier raw decision values vs probability estimates, binary-class training vs multi-class training to only name a few). It remains unclear how these, and especially their combinations, impact TDLM's ability to differentiate true sequenceness results from a systemic bias due to, for example, endogenous oscillatory confounders. To improve methodological certainty, further studies could usefully investigate this question in the form of a multiverse re-analysis of previously published data (*Chambers, 2020*; *Dafflon et al., 2022*).

## Improving TDLM sensitivity

TDLM is a complex and sophisticated approach for detecting human replay. However, in our study, TDLM was not sufficiently sensitive to detect replay during offline periods, under an assumption that replay was indeed present in this 8-min rest period. Nonetheless, we suggest several factors could potentially be improved to increase its power and sensitivity. First, the success of the approach hinges on the quality of the decoders. Improving their specificity as well as their ability to transfer from the localizer to a condition of interest should be assessed more closely. While most decoding approaches simply assume that transfer is seamless, previous research has indicated that this depends heavily on target modality, such as the way the decoders are trained as well as the cognitive processes involved (*Bezsudnova et al., 2024*; *Dijkstra et al., 2019*; *Dijkstra et al., 2021*). Further research should optimize and validate that decoders trained primarily on sensory evoked activity can be flexibly applied to replay detection, for example by introducing a control that demonstrates classification accuracy in relation to another sensory modality. If transfer is not possible from visual sensory to, for example auditory or imagined stimuli, it might indicate failure of representational transfer that is likely to be a necessity for detection of reactivation during replay. Relatedly, focusing on the peak of the decodability probably introduces a reliance on sensory processing that does not necessarily transfer well to other, sensory and non-sensory, conditions. Thus, training decoders independently, or as an ensemble of the time dimension, as, for example, performed by *Eldar et al., 2020* or *Wise et al., 2021* could help circumvent this problem. Additionally, a wide variety of oddballs has been used (e.g. upside-down, scrambled, or mismatched images, cues presented visually, as words, auditorily, etc), and at this time, it is unclear if these affect the representations that the classifier learns. Similarly, performing localizers on categorical stimuli instead of specific stimuli, as well as training on multi-sensory localizers, might prove fruitful, as it reduces overreliance on specific visual features that might not be central to the reactivated memory representation. In summary, we would expect a multimodal categorical localizer and a classifier that is not trained on a specific timepoint to generalize best.

Second, as our simulation shows, detecting replay in shorter windows of time should be possible, as here a high replay density needed has previously been reported (*Norman et al., 2019*). Therefore, restricting the analysis to time 'regions of interest' wherein a high replay frequency is expected could drastically improve detection power. As it is known that replay occurs in bursts, such windows might exceed necessary replay densities. At this stage, we cannot offer a general recommendation for window sizes as they are likely to depend on details of the research paradigm. However, intracranial recordings can be used as a proxy to estimate the duration of replay bursts, for example as reported in *Norman et al., 2019* increased SWRs were seen up to 1500 ms after retrieval cue onset. While studies researching planning or retrieval naturally offer such restricted analysis windows, this becomes more difficult for longer recordings such as resting state and sleep analysis. Furthermore, to prevent false positives and 'double-dipping' (*Kriegeskorte et al., 2009*), such time-windows would need to be chosen beforehand or include an appropriate control for multiple comparisons.

Detection might also be enhanced through use of contrasts of specified oscillation phases (e.g. slow oscillations during sleep or theta phases during resting state) during which it is known that replay typically occurs (*Xiao et al., 2024*). Another approach would be contrasting windows after targeted memory reactivation cues and respective control phases of control stimulation to validate the replay detection (*Chen et al., 2024*). Last but not least, search windows could also be artificially narrowed by using a sliding window similar to the approach implemented by *Wise et al., 2021*. However, in this case, multiple comparison problems between different windows are introduced, and circular testing would need to be avoided (e.g. by using exploration and validation datasets). Nevertheless, as the considerable branching factor poses a threat of increased false-positive findings, we opt to focus the current simulations on previously published pipelines and parameters. Future studies should systematically evaluate parameter choices on TDLM under different conditions, something beyond the remit of the current study.

Lastly, there are certain assumptions that TDLM makes that might not hold (see Materials and methods Study II). Current implementations look for a fixed time lag that is the same across all participants and between all reactivation events. If time lags differ across participants, TDLM will fail to find them. Similarly, TDLM assumes a fixed sequence order and is not robust against slight within-sequence permutations or in-sequence-missing reactivation events. However, from other data sources, such as hippocampal place cell recordings, it is known that such permutations can occur where some states are skipped or fail to decode during replay. Similarly, it is assumed that each reactivation event lasts between 10 and 30 ms, but the true temporal evolution of reactivation measured by TDLM is currently unknown. Future method development might focus on improving invariance to these assumptions.

Furthermore, in this paper, in collaboration with some of the original authors of TDLM, we introduced a novel way to test for significance that is more robust and sensitive than that previously used state-permutation maximum. The newly proposed sign-flip permutation test has markedly increased power and is more robust, as can be observed from the analysis presented in *Figure 6—figure supplement 1*, and its use is strongly recommended for future studies using TDLM.

Finally, what do our simulations imply for the broader MEG replay literature? Our implementation successfully detects replay when boundary conditions are met, as shown in the simulation. But sensitivity depends critically on high fidelity between the analysis window and the density of replay events. A systematic evaluation of these conditions as they apply to prior studies remains beyond the scope of the current paper. Instead, our focus is on delineating boundary conditions that we hope will motivate conduct of power analyses in future work as well as inclusion of simulations that approximate realistic experimental conditions.

In summary, there are several areas where TDLM might be improved, including a restriction in its search space, improvement in classifiers, a validation of localizer representation transfer to other domains (e.g. memory representations), and the extension of TDLM to render it more robust against violations of its core assumptions.

## Limitations

In an effort to explain null findings, we conducted a simulation study. Despite close attention to the bounds of this simulation, it is possible that the reported results are unique to our analysis pipeline or recording device (e.g. application of MaxFilter, see Materials and methods section), even though we rigorously followed best practices of the field. While we are reasonably confident the proposed

approach approximates expected veridical replay situations, certain assumptions of our analysis may be too strong and not model actual replay well (e.g. the way we chose to extract and insert neural patterns). Furthermore, it is possible that TDLM behaves differently with different MEG systems, different paradigms, pre-processing, or different analysis approaches and that we simply did not find the correct parameters to make it work. Nevertheless, we closely followed previous pipelines applied on the same dataset (*Kern et al., 2024*) that did find effects using TDLM and also tried many plausible alternative pipelines without success. Additionally, while the analysis we propose arguably simulates replay under more realistic conditions than previous work, there are still ways to increase realism even further that were out-of-scope for this paper. Future studies could, for example simulate nesting of replay with oscillations such as gamma or theta or look at how clustered replay affects sequenceness. Finally, while initially planning for thirty participants, due to exclusion criteria, our study featured fewer participants than most previous studies using TDLM (i.e. usually 25–40, but 21 in our study). While we are confident that our simulation results hold under these sample sizes, as sample sizes of other studies show comparable power to our (*Figure 6—figure supplement 1*), we cannot rule out a possibility that our null findings are explained by a lack in power alone.

## Conclusion

We were unable to detect replay during an 8-min rest epoch in our data set using TDLM. Conducting a simulation, we first show that even if replay was present, implausibly high replay densities would be necessary to achieve significance at our sample size. Second, despite our best efforts, we were unable to induce a correlation of sequenceness with behavior at the individual level and uncovered large baseline fluctuations in sequenceness across participants. Taken together, our results suggest that future studies should refine and validate decoder training paradigms and limit the search space for successful replay detection during resting states via TDLM. Additionally, future studies should calculate expected replay densities that would be necessary for reported results to demonstrate plausibility. Finally, our results highlight the overall importance of delineating the limits of replay detection for the interpretation of null effects.

## Materials and methods

### Material and methods study I: resting state analysis

#### Participants

Note that parts of this methods section are taken from *Kern et al., 2024* for convenience of the reader. We recruited thirty participants (15 male and 15 female-identifying), between 19 and 32 years old (mean age 24.7 years). Inclusion criteria were right-handedness, no claustrophobic tendencies, no current or previous diagnosed mental disorder, non-smoker, fluency in German or English, age between 18 and 35, and normal or corrected-to-normal vision. Caffeine intake was requested to be restricted for four hours before the experiment. Participants were recruited through the institute's website and mailing list and various local Facebook groups. A single participant was excluded due to a corrupted data file and replaced with another participant. We acquired written informed consent from all participants, including consent to share anonymized raw and processed data in an open access online repository. The study protocol was approved by the ethics committee of the Medical Faculty Mannheim of Heidelberg University (ID: 2020–609). The experiment and analysis were preregistered at aspredicted.org (https://aspredicted.org/kx9xh.pdf, #68915).

#### Procedure

Participants came to the laboratory for a single study session of approximately 2.5 hr. After filling out a questionnaire about their general health, their subjective sleepiness (Stanford Sleepiness Scale, *Hoddes et al., 1973*) and mood (PANAS, *Watson et al., 1988*), participants performed five separate tasks while in the MEG scanner. First, before any stimulus exposure, an 8-min eyes-closed resting state was recorded. This was followed by a localizer task (~30 min), in which all 10 items were presented 54 times in pseudo-randomized order, using auditory and visual stimuli. Next, participants learned a sequence of the 10 visual items embedded into a graph structure until they achieved 80% accuracy or reached a maximum of six blocks (7–20 min). Following this, we recorded another 8-min eyes-closed resting state to allow for initial consolidation and, finally, a cued retrieval session (4 min), of which

the results have previously been reported in *Kern et al., 2024*. For an overview, see *Figure 1*. For an animated version of the task, see GitHub pages here. Note: For brevity, we have only included information that is relevant for the understanding of the current study. Further details on the localizer, learning, and retrieval task as well as the stimulus material can be found in *Kern et al., 2024*.

### Localizer task

Ten stimuli were shown to participants repeatedly in a pseudo-randomized order with 54 presentations per image. They were images drawn from *Rossion and Pourtois, 2001*, selected to be diverse in color, shape, and category and for having short descriptive words (one or two syllables) both in German and English. In each trial, a fixation cross was shown, and after 0.75–1.25 s a word describing the current stimulus was played via headphones. Auditory stimuli were created using the Google text-to-speech API, availing of the default male voice (SsmlVoiceGender.NEUTRAL) with the image description labels, either in German or English, based on the participants' language preference. Auditory stimulus length ranged from 0.66 to 0.95 s. Next, the stimulus image was shown for 1 s. As a distractor, in ~11% of the trials, the audio and image were incongruent, which participants should indicate with a button press. These oddball trials were excluded from all further analysis and decoder training. Image transitions were balanced such that each image was followed by each other image the same number of times.

### Graph learning

The ten images were randomly assigned to nodes of the graph, as shown in *Figure 10B*. Participants were instructed to learn a randomized sequence of elements with the goal of reaching 80% performance within six blocks of learning. During each block, participants were presented with each of the 12 edges of the graph exactly once.

After a fixation cross of 3.0 s, a first image was shown on the left of the screen. After 1.5 s, the second image appeared in the middle of the screen. After another 1.5 s, three possible choices were displayed in vertical order to the right of the two other images. One of the three choice options was the correct successor of the cued edge. Of the two distractor stimuli, one was chosen from a distal location on the graph (five to eight positions away from the current item), and one was chosen from a close location (two to four positions away from the current item), and neither was connected to the currently onscreen item on the graph. Feedback was shown for 3 s. No audio was played during learning. After completing the task, participants were told that their knowledge would be prompted once more a few minutes later (see retrieval).

### Resting state

Two resting state recordings were recorded in the MEG. One before any stimulus exposure at the beginning of the experiment (pre-RS), as well as a second one (post-RS) after graph learning, before the retrieval part. Each resting state session lasted 8 min. Participants were instructed to close their eyes and to 'not think of anything particular'.

### Retrieval

After the resting state, we presented participants with a single retrieval session block, which followed the exact layout of the learning task with the exception that no feedback was provided as to whether entered choices were correct or incorrect (*Figure 10D*). Analysis of the retrieval period and more methodological details can be found in *Kern et al., 2024*.

### MEG acquisition and pre-processing

MEG was recorded in a passively shielded room with a MEGIN TRIUX (MEGIN Oy, Helsinki, Finland) with 306 sensors (204 planar gradiometers and 102 magnetometers) at 1000 Hz with a 0.1–330 Hz band-pass acquisition filter at the ZIPP facility of the Central Institute for Mental Health in Mannheim, Germany. Before each recording, empty room measurements ensured no ill-functioning sensors were present. Head movement was recorded using five head positioning coils. Bipolar vertical and horizontal electrooculography (EOG) as well as electrocardiography (ECG) was recorded. After recording, the MEGIN proprietary MaxFilter algorithm (version 2.2.14) was run using temporally extended signal

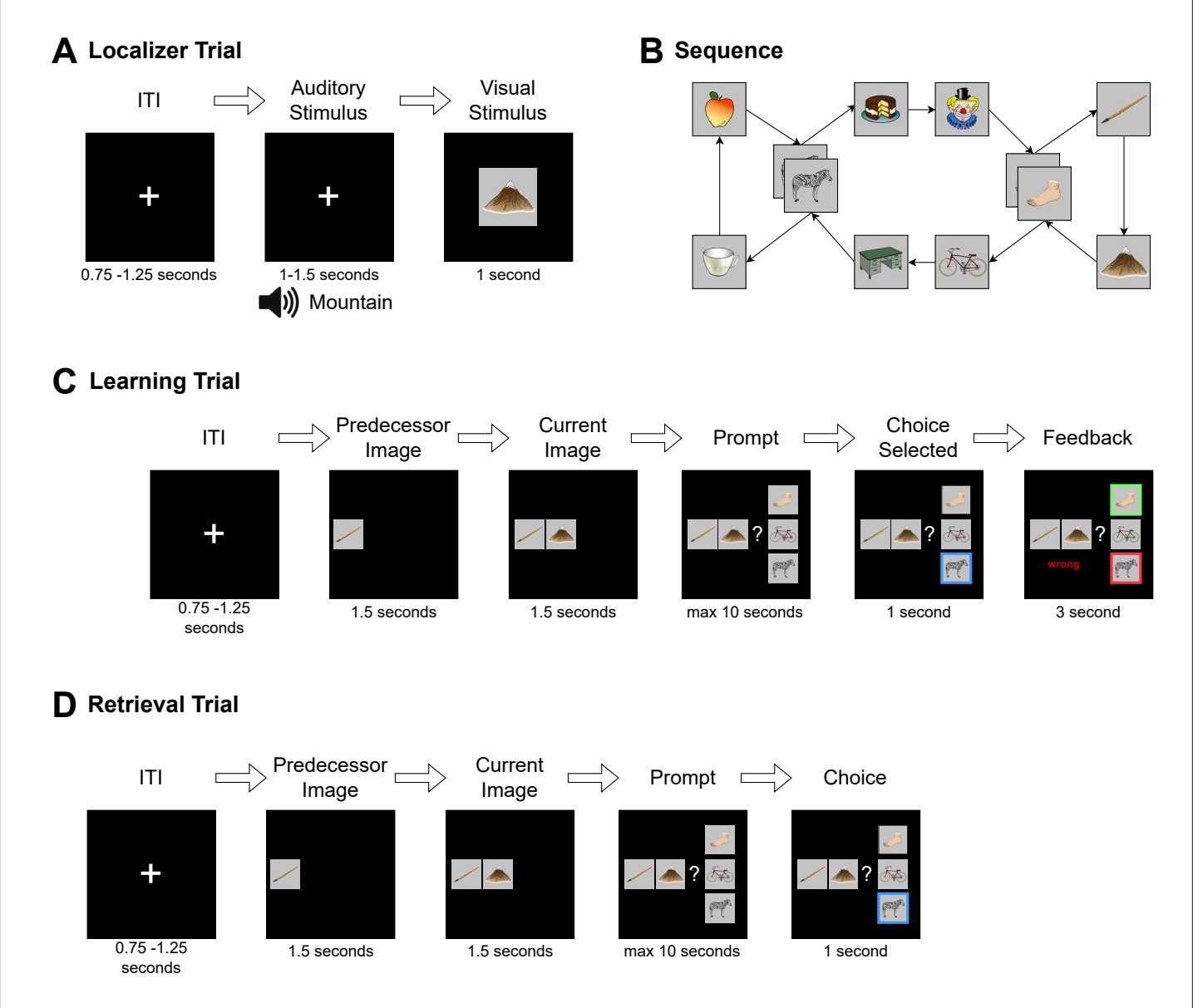

**Figure 10.** Detailed overview of the experiment procedure. (**A**) During the localizer task, a word describing the stimulus was played via headphones and the corresponding visual item was then shown to the participant. In 11% of trials, the audio and visual cue did not match, and in this case, participants were instructed to press a button on detection (to sustain attention to the stimuli and check for inattention). (**B**) Graph layout of the task. Two elements could appear in two different triplets. The graph was directed such that each tuple had exactly one successor (e.g. apple→zebra could only be followed by cake and not mug), but individual items could have different successors (zebra alone could be followed by mug or cake). Participants never saw the illustrated birds-eye view. (**C**) During learning, in each trial, one node was randomly chosen as the current node. First, its predecessor node was shown, followed by the current node with the participant then given a choice of three items. They were then required to choose the node that followed the displayed cue tuple. Feedback was then provided to the participant. This process was repeated until the participant reached 80% accuracy for any block or reached a maximum of six blocks of learning. (**D**) The retrieval followed the same structure as the learning task, except that no feedback was given.

space separation (tSSS) and movement correction with the MaxFilter default parameters (***Taulu and Simola, 2006***, raw data buffer length of 10 s, and a subspace correlation limit of .98. Bad channels were automatically detected at a detection limit of 7; none had to be excluded. The head movement correction algorithm used 200ms windows and steps of 10ms. The HPI coil fit accept limits were set at an error of 5 mm and a g-value of 0.98). Using the head movement correction algorithm, the signals were virtually re-positioned to the mean head position during the initial localizer task to ensure compatibility of sensor-level analysis across the recording blocks. The systematic trigger delay of our

presentation system was measured, and visual stimuli appeared consistently 19 ms after their trigger value was written to the stimulus channel. However, to keep consistency with previous studies that do not report trigger delay, timings in this publication are reported uncorrected (i.e. 'as is', not corrected for this delay).

Trial data were pre-processed using Python-MNE (version 1.7, RRID:SCR_005972, *Gramfort et al., 2013*). Data were down-sampled to 100 Hz using the MNE function 'resample' (with default settings, which applies an anti-aliasing filter before resampling with a brick-wall filter at the Nyquist frequency in the frequency domain) and ICA applied using the 'picard' algorithm (*Ablin et al., 2018*) on a 1 Hz high-pass filtered copy of the signal using 50 components. As recommended, ICA was set to ignore segments that were marked as bad by Autoreject (*Jas et al., 2017*, RRID:SCR_022515) on 2-s segments. Components belonging to EOG or ECG and muscle artifacts were identified and removed automatically using MNE functions 'find_bads_eog', 'find_bads_ecg' and 'find_bads_emg', using the EOG and ECG as reference signals. Finally, to reduce noise and drift, data were filtered with a high-pass filter of 0.5 Hz using the MNE filter default settings (hamming window FIR filter, –6 dB cutoff at 0.25 Hz, 53 dB stop-band attenuation, filter length 6.6 s). To improve numerical stability, signals were re-scaled to similar ranges by multiplying values from gradiometers by 1e10 and from magnetometers by 2e11. These values were chosen empirically by matching histograms for both channel types. As outlier values can have significant influence on the computations, after re-scaling, values that were still above 1 or below –1 were 'cut off' and transformed to smaller values by multiplying with 1e-2.

Trials from the localizer task were created from –0.1 to 0.5 s relative to visual stimulus onset to train decoders. We realized that the initial pre-registered window of 0–0.3 s was rather short, and we deviated from the pre-registration by extending the analysis window by 100 ms before and 200 ms after stimulus onset. However, the deviation was only to improve interpretability and to include baseline data and did not alter the results in any way. No baseline correction was applied. To detect artifacts, Autoreject was applied using default settings, which repaired segments by interpolation in case artifacts were present in only a limited number of channels and rejected trials otherwise. Anonymized and max-filtered raw data are openly available at Zenodo (RRID:SCR_004129), the localizer data has been previously published as part of a previous publication (v1 https://doi.org/10.5281/zenodo.8001755), the resting state data can be found at v2 (https://doi.org/10.5281/zenodo.15629081), code and behavioral data will be made public on GitHub (RRID:SCR_002630) here (copy archived at *Kern, 2026*).*Figure 5*

From the resting state, segments were taken between the onset and offset markers, which were approximately 8 min apart and in which the participant had the eyes closed. 'Autoreject' is not well tested on longer segment data; therefore, unlike for the localizer trials, we did not apply it to the resting state data. However, a supplementary analysis revealed that both ICA and Autoreject did not have any significant effect on decoding accuracy, a result also reported in other studies (*Delorme, 2022*).

## Split in exploration and validation dataset

As stated in the pre-registration, we used 50% of the resting state data for exploration to tune classifiers and try out different approaches. To mitigate time effects on replay, we used interleaved 30-s segments of the resting state data to divide into exploration and validation sets. To counterbalance ordering effects, for the first 15 participants, we concatenated the uneven segments as exploration, while for the latter participants, we switched the assignment. After settling for the parameters reported in this section, we used the other 50% of data to validate the results. All results described in Study I are from the validation segments.

## Decoding framework and training

We closely followed the approaches applied in previous research (*Kurth-Nelson et al., 2016*; *Liu et al., 2019*; *Wimmer et al., 2020*) and used the same data set as reported in our previous study (*Kern et al., 2024*). We used Lasso regularized logistic regression on sensor-level data of localizer trials using the Python package Scikit-Learn (*Pedregosa et al., 2011*, RRID:SCR_002577). Decoders were trained separately for each participant and each stimulus, using liblinear as a solver with 1000 maximum iterations and a L1 regularization of C=9.1, which was determined by cross-validating the decoding accuracy in the time span of 150–250 ms after stimulus onset across participants using

values for C of a log-spaced interval from $10^{-2}$ to $10^2$ (see *Figure 2—figure supplement 1* for details). To circumvent class imbalance due to trials removed by Autoreject, localizer trials were stratified such that they contained an equal number of trials from each stimulus presentation by randomly removing trials from over-represented classes. Using a cross-validation schema (leaving one trial out for each stimulus per fold, i.e. 10 trials left out per fold), for each participant, the decoding accuracy was determined across time (*Figure 2B*). During cross-validation, for each fold, decoders were trained on data of each 10 ms time step and tested on left-out data from the same time step. Therefore, decoding accuracy reflects the separability of the stimulus classes by the sensor values for each time step independently. Decoders were trained using a one-vs-all approach, which means that for each class, a separate classifier was trained using positive examples (target class) and negative examples (all other classes) plus null examples (data from before stimulus presentation, see below). This approach allows the decoder to provide independent estimates of detected events for each class.

For each participant, a final set of decoders (i.e. 10 decoders per participant, for each stimulus one decoder) was trained at 210 ms after stimulus onset, a time point reflecting the average peak decoding time point computed for all participants. For the final decoders, data from before the auditory stimulus onset was added as a negative class with a ratio of 1:2, based upon results from previous publications reaching better sensitivity with higher negative class ratio (*Liu et al., 2021a*). Adding null data allows decoders to report low probabilities for all classes simultaneously in absence of a matching signal and reduces false positives while retaining relative probabilities between true classes. Together with the use of a sparsity constraint on the logistic regression coefficients, this increases the sensitivity of sequence detection by reducing spatial correlations of decoder weights (see also *Liu et al., 2021a*). For a visualization of relevant sensor positions, see the Supplementary section of *Kern et al., 2024*. Decoders were then applied to the two-resting state session data. This resulted in ten probability vectors across the resting state sessions, one for each item, in steps of 10 ms. These probabilities indicate the similarity of the current sensor-level activity to the activity pattern elicited by exposure to the stimulus and can therefore be used as a proxy for detecting active representations, akin to a representational pattern analysis approach (RSA, *Grootswagers et al., 2017*).

## Sequential replay analysis

To test whether individual items were reactivated in sequence at a particular time lag, we applied TDLM (*Liu et al., 2021b*) to the resting state segments, using a Python (RRID:SCR_008394) implementation of the algorithm (TDLM-Python, *Kern, 2024*). In brief, the TDLM method approximates a time-lagged cross-correlation of reactivation strength in the context of a particular transition pattern, quantifying evidence for a certain activity transition pattern distributed in time. As input for the sequential analysis, we used the probabilities of the ten classifiers corresponding to the stimuli, which we normalized by dividing each class probability by the mean of all classes.

Using a linear model, the method first estimates evidence for sequential activation of the decoded item representations at different time lags. For each item i at each time lag Δt up to 300 ms it estimates a linear model of form:

$$Y\_i = Y\left(\Delta t\right) \times \beta\_i\left(\Delta t\right)$$

where Y_i contains the decoded probability output of the classifier of item i and Y(Δt) is simply Y time lagged by Δt. When solving this equation for β_i (Δt), the predictive strength of Y(Δt) for the occurrence of Y_i at each time lag Δt can be estimated. Calculated for each stimulus i, an empirical transition matrix T_e (Δt) that indexes evidence for a transition of any item j to item i at time lag Δt can be created (i.e. a 10x10 transition matrix per time lag, each column j contains the predictive strength of j for each item i at time lag Δt). These matrices are then combined with a ground truth transition matrix T (encoding the valid sequence transitions of interest) by taking the Frobenius inner product. This returns a single value Z_Δt for each time lag, indicating how strongly the detected transitions in the empirical data follow expected task transitions, which is termed 'sequenceness'. Using different transition matrices to depict forward (T_f) and backward (T_b) replay, evidence for replay at different time lags for each trial is estimated separately. To control for auto-correlations of the signal, caused by large brain oscillations such as occipital alpha (*Liu et al., 2021a*), we added time-shifted versions of the decoded probabilities to the GLM in 10 Hz intervals. This process is applied to each trial individually, and resulting sequenceness values are averaged

to provide a final sequenceness value per participant for each time lag Δt. To test for statistical significance, a baseline distribution is sampled by permuting the rows of the transition matrix 1000 times, creating transition matrices with random transitions; identity-based permutation (*Liu et al., 2021a*) and calculating sequenceness across all time lags for each permutation. The null distribution is then constructed by taking the peak sequenceness across all time lags for each permutation. To minimize possible participant-specific biases on the magnitude of sequenceness values, we z-scored along the time lag axis for each permutation, for each participant, similar to previously proposed (*Wimmer et al., 2020*).

## Sign-flip-permutation test

As previously noted (*Liu et al., 2021a*), the standard approach of taking the maximum or 95th percentile of sequenceness from state-shuffled permutations controls for multiple comparisons very conservatively. Its method implements a fixed-effects model, as it averages data across participants to establish a threshold without accounting for inter-subject variability. Consequently, the results are susceptible to outliers, which we partially tried to mitigate by z-scoring sequenceness values. To implement a random-effects inference framework that accounts for between-participant variance, a sign-flip permutation test can be employed. This approach is significantly more robust to outliers and follows established non-parametric methods for multiple comparison correction in neuroimaging (*Nichols and Holmes, 2002*). In this procedure, the sign of the entire sequenceness curve for a random subset of participants is inverted for each permutation. A one-sided t-test against zero is then calculated at every time lag for each permutation. By taking the maximum t-value across all time lags for each permutation, an empirical null distribution of the maximum statistic is constructed. The observed t-values are then compared against this distribution, providing Family-Wise Error Rate (FWER) control across the temporal axis. This method offers higher sensitivity than state-shuffling because it preserves the temporal structure of the individual data while explicitly penalizing effects that are inconsistent across the cohort. Furthermore, this framework naturally extends to more complex experimental designs, such as multiverse analyses involving varying classifier parameters or regularization strengths.

## Resting state analysis

To assess whether previously learned sequences are replayed during the resting state, we tested for significant sequenceness using the above-mentioned permutation test for forward and backward sequenceness. We tested for all allowable directed graph transitions. Additionally, we compared the sequenceness results to the control resting state by subtracting the sequenceness values on the participant level. Additionally to the maximum across all permutations, we report the 95% confidence interval of the maxima across each permutation (*Liu et al., 2021a*). Finally, to assess if sequential replay was correlated with the participant's performance, we calculated a Pearson correlation between peak sequenceness across all time lags and the retrieval performance (as in *Kern et al., 2024*).

## Parameter exploration

For completeness and transparency, we report what parameters we explored during the exploration phase on the first half of the data set (final results were calculated on a held-out portion of the resting state data). For classifiers, we tried Random Forest Classifiers, Support-Vector Machines as well as LDA. We applied different normalizations of the MEG data by z-scoring the raw data along channels, times as well as across the entire recording. These were all inferior in terms of localizer decoding accuracy compared to the current approach. We trained on single time points and averaged time windows, including PCA. We employed z-scoring on the sequenceness values, on the probability estimates as well as using raw classifier outputs instead of probability estimates or normalizing each class by its mean probability. Nevertheless, after considering the vast space of possible parameters, we did not find alternative parameters that robustly showed a predicted replay pattern and thus, to avoid overfitting and remain comparable to the literature, we report this study with parameters most often used in previous research in terms of normalization, classifier choice (Logistic Regression) as well as training regimen. Crucially, these parameters were successfully applied by us to identify replay effects in our previous study using the retrieval data from this data set (*Kern et al., 2024*).

## Exclusions

Replay analysis relies on a successive detection of stimuli, where the chance of detection exponentially decreases with each step (e.g. detecting two successive stimuli with a chance of 30% leaves a 9% chance of detecting a replay event). However, one needs to bear in mind that accuracy is a 'winner-takes-all' metric indicating whether the top choice also has the highest probability, disregarding subtle, relative changes in assigned probability. As the methods used in this analysis are performed on probability estimates, and not class labels, one can expect that 30% is a rough lower bound and that the actual sensitivity within the analysis is higher. Additionally, based on pilot data, we found that attentive participants were able to reach 30% decodability, allowing its use as a data quality check. Therefore, we decided a priori that participants with a peak decoding accuracy of below 30% would be excluded from the analysis (nine participants in total), as obtained from the cross-validation of localizer trials. Additionally, as successful above-chance learning was necessary for the paradigm, we ensured all remaining participants had a retrieval performance of at least 50% (one participant had to be excluded, but was already excluded due to low decoding performance). No participant had more than 25% missed button presses, so no exclusion was necessary on this pre-registered criterion.

## Material and methods study II: simulation

To better understand our null finding, we conducted a simulation study. We applied a hybrid simulation approach, using resting state data recorded before any stimulus exposure as a sequence-free baseline, into which we inserted neural patterns taken from the localizer. By inserting these patterns at a specific time lag, we can create a ground truth of replay-like events that should be well-decodable. The advantage of using real-world MEG data over synthetic data is that it includes all nuisance confounders that might be present under realistic conditions, yet leaves room to control the variable of interest. Finally, by varying the density and length of inserted events, we can assess an upper bound of sensitivity of TDLM under optimal, yet realistic conditions.

## Simulating replay by inserting localizer data into the control resting state

TDLM makes assumptions (implicit and explicit) about the underlying structure of replay. While it has yet to be shown these assumptions are based on the characteristics of actual replay, all methods pose other constraints (*Cliff et al., 2023*). Here, we aimed to create an optimal scenario tailored to TDLM. Therefore, we adhere to the following assumptions:

1. Individual items are reactivated and can be decoded throughout the recording.
2. Individual item reactivations are brief, usually only of ~10–30 ms (as for example shown in the replay example visualization in Figure 2 in *Liu et al., 2019* or Figure 3 in *Wimmer et al., 2020*) duration.
3. Item reactivations are non-random and adhere to a specific sequential pattern.
4. A replay event comprises two or more reactivation events in close temporal proximity.
5. There is a fixed time lag between all reactivations across all replay events

Additionally, we include the following constraints for the simulation:

1. Classifiers trained on the peak time point of decodability of sensors from a localizer transfer well to reactivation events
2. Between each replay event (two or more stimulus reactivations), there is a refractory period of at least the time lag of interest in the analysis (i.e. 150 ms)
3. Replay is correlated with memory performance during the retrieval session.
4. The replay density is stable across the recording.

Based on these assumptions, we inserted replay events into the control resting state of each participant by first extracting stimulus-specific patterns of sensor activity and then adding these patterns at sequential positions in the resting state. We created patterns by taking the mean sensor activation at the peak of decodability during the localizer of a specific class (as defined above, i.e. at 210 ms). To isolate the stimulus-specific pattern from activity elicited by general sensory processing, we then subtracted the mean sensor values across all other classes (akin to an event-related potential of seeing any stimulus except the current target stimulus). For a visualization of the mean patterns across participants, see *Figure 5—figure supplement 1*. Next, we determined N replay onset points *t* throughout the resting state and inserted the pattern of a randomly chosen class by simply adding the pattern to

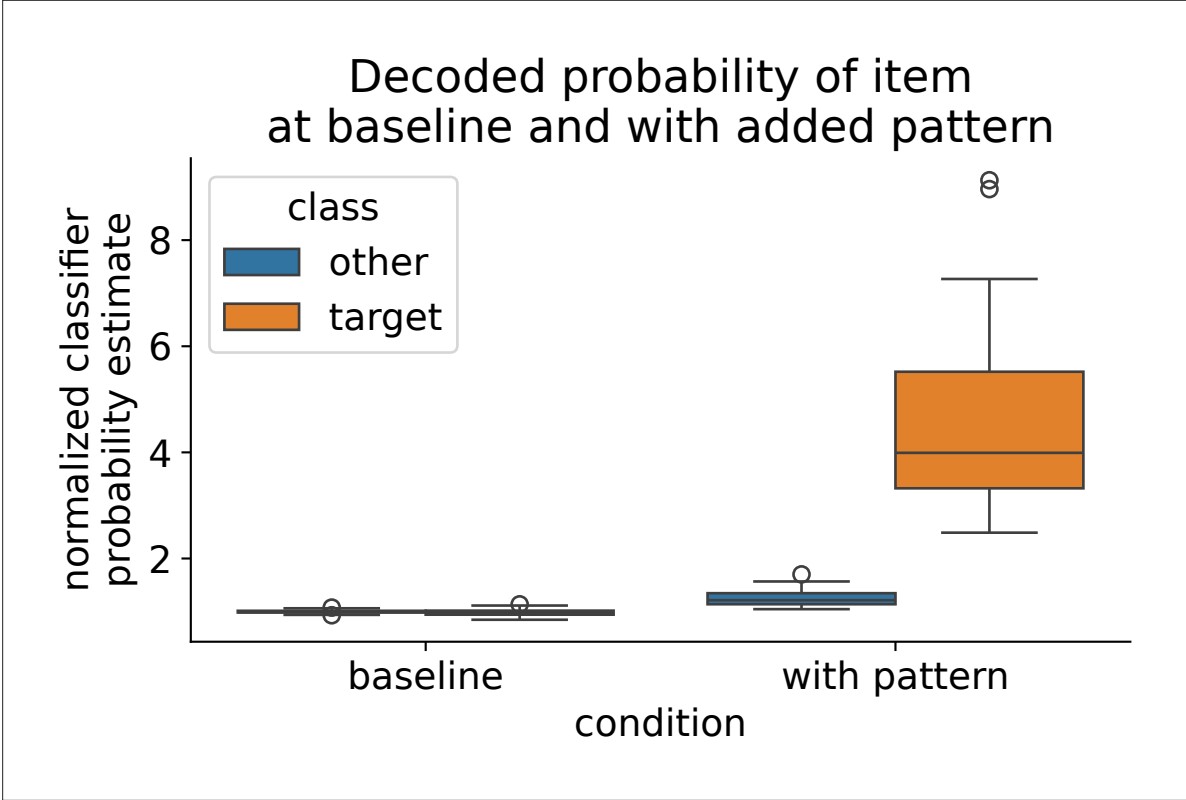

**Figure 11.** Normalized probability estimates of time points in the control resting state for the target class of which the pattern has been added and non-target class. Left: Time points at which no pattern has been inserted, both classes have a similar probability estimate at baseline. Right: At time points where the stimulus-specific pattern was added, the decoded probability estimate is significantly increased. Note that also the probability estimate of non-target classes is slightly increased at the timepoint of pattern insertion, which is due to slight shifts in sensor values that heighten all probabilities by a small amount. For the mean classifier probabilities before and after normalization by dividing by the mean probability per class, see *Figure 11—figure supplement 1*. Visualization of the sensor values before and after insertion shows that core distribution characteristics of the signal are retained, see *Figure 11—figure supplement 2*.

The online version of this article includes the following figure supplement(s) for figure 11:

**Figure supplement 1.** Mean classifier probabilities before and after normalization, per participant for time points selected for replay pattern insertion.

**Figure supplement 2.** Distribution of raw sensor values before and after insertion of the simulation pattern.

the sensor values at time $t_i$. After a fixed time lag of 80 ms after $t_i$, we inserted the pattern of the class that would follow according to the underlying sequence structure. This time lag (80 ms) was chosen in order to isolate precisely an effect of the experimentally inserted sequenceness. Thus, we chose a lag at which the mean baseline sequenceness was close to zero and where the correlation with behavior was low. Additionally, we subtracted the mean sequenceness value (at 80 ms) at baseline from the specific lag recorded for each participant, such that simulation effects would be initialized at zero sequenceness on average enabling any effects to be attributed purely to inserted replay. Additionally, we excluded time lags too close to the alpha rhythms of participants (which varied between ~9 and 11 Hz) or lags which would have a harmonic with the rhythm. We did not allow 150 ms after and before each replay event from containing any other replay events, as this would have introduced interactions between two replay events within the used search window of 300 ms. For simplicity, we only used one-step replay event transitions, that is containing two reactivation events.

*Figure 11* shows the decoded probabilities at the time points $t$ before and after addition of the stimulus-specific pattern. The target class is now well decodable, while other classes remain at a low baseline probability (*Figure 11*). Using this approach, we can simulate replay events while for the most part preserving other characteristics of the resting state in terms of endogenous oscillations that might introduce spurious sequenceness. However, note that this simulation approach is likely to slightly overestimate decodability of reactivation events (which reflects in an average decoding

accuracy of 43% after insertion which is higher than the decoding accuracy during the localizer). Additionally, as the classifier follows a one-versus-all classification approach, it estimates probabilities for each class separately. Therefore, a small increase in probability can be seen even for non-target classes, which most probably stems from remaining patterns from general sensory processing and the slight increase in sensor magnitude after addition. Sensor values still follow the distribution of a naturalistic MEG signal after inserting patterns, that is are centered around zero and have a comparable standard deviation (see *Figure 11—figure supplement 2*).

## Which pattern to extract from the localizer

To extract a suitable pattern that can be added to the control resting state, some constraints need to be addressed. First, the pattern should elicit an increase in probability specific to the target class when added to the resting state data. Consequently, all non-target classes should not increase significantly compared to baseline. Because if the baseline probability is increased, spurious sequenceness will occur simply due to the baseline increase, as is the case with oscillatory confounders (*Liu et al., 2021a*). Additionally, classifiers will also react to any event-locked visual activity, slightly increasing the probability of the target class, even if seeing an unrelated image. We initially elicited these patterns by taking the mean ERP per class and adding the sensor values of the peak decoding time point to resting state positions. However, this shifted the distribution of sensor values in such a way that the baseline probability of non-target classes also increased (*Figure 11*). Therefore, we opted to refine the pattern by taking the ERP of all other classes and subtracting it from the ERP of the target class. This has the advantage that it removes most of the background visual activity and only keeps activity specific to the target class.

## Replay density

Unlike intracranial or cell recordings, TDLM does not per se detect individual replay events, but given a specific transition pattern ascertains evidence for sequenceness over a period of time. Each in-sequence reactivation of two or more items increases the specific time lag's sequenceness value. Therefore, the sequenceness magnitude strongly relies on the underlying density of decoded replay events, which is usually denoted in events per minute or $min^{-1}$. To assess the sensitivity of TDLM, we incrementally increased the density up to a theoretical maximum of around 200 $min^{-1}$, as given by the time lag of 160 ms plus the above-defined refractory period of 150 ms. Additionally, as replay is seen as instrumental to memory consolidation, the density was linearly scaled with the participant's retrieval memory performance (e.g. a participant with 50% memory performance had at maximum 80 $min^{-1}$=160 $min^{-1}$ * 50% events inserted). To assess how performance might affect replay in our specific dataset, we chose to use the original participants' performance values instead of uniformly sampling the performance space (which ranged from 50% to 100%). However, for the correlation analysis, we additionally added a 'best-case' scenario, in which we scale replay from 0 to 1, an approach that is statistically equivalent to scaling values to the full space of possible performance (0 to 100%; see Results Study II: Simulation). Note that we deliberately overestimated the link between replay and memory performance, which has been estimated to be as low as $r$=0.2 in humans (*Schapiro et al., 2018*).

Additionally, the time points of inserting replay within the resting state are sampled from a uniform distribution. Even though TDLM tracks reactivation events over time, at a macro scale, the algorithm is invariant to the temporal distribution. At each time step, the GLM regresses onto a future time step up to the maximum time lag of interest, yielding a predictor per lag. However, these predictors within the GLM are independently assessed, and hence, TDLM is, outside of the time lag window, relatively invariant to the temporal distribution of replay. To demonstrate our claim, we simulated uniform replay vs 'bursty' replay that only occurs in some parts of the resting state; both yield equivalent sequenceness results (see *Appendix 1—figure 1* for details).

## Reactivation event length

It is unclear how long the span of individual decodable reactivation events is in the context of MEG decoding. In primates, sharp wave ripples (SWR), as a detectable proxy for a reactivation event, are usually searched for in the range of 20–200 ms (*Liu et al., 2022a*), while mostly found to range from 50 to 100 ms (*Buzsáki, 2015*), hence each individual reactivation is much shorter. There is no report

of decoded reactivation event lengths in the context of MEG; however, spans can be estimated from example illustrations given by previous studies, where they mostly span 10–30 ms. Nevertheless, even if reactivation events are brief, there seems to be a waxing period, a peak, and a waning period. Therefore, we simulated this process by extracting the item-specific pattern around the decoding accuracy peak and weighting it with a Gaussian (sigma = 1, i.e. item-specific pattern at 210 ms ± 20 ms was inserted into 50 ms centered around the reactivation event marker in the resting state with the following weighting: [0.058, 0.24, 1, 0.24, 0.058]).

## Synthetic simulation

In our paper, we argue that a hybrid simulation is superior to previously proposed purely synthetic simulation approaches, mainly for overestimated pattern discriminability of synthetic data as well as a lack of oscillatory confounds. We have conducted a purely synthetic simulation that is based upon the sample code proposed in *Liu et al., 2021a*. Following their sample code, we simulated baseline resting state MEG by first sampling a symmetric covariance matrix to simulate sensor dependencies. Using this covariance matrix, we generated spatially correlated noise and introduced a temporal dependency by ensuring each time point was strongly correlated with the previous one (correlation of 0.95, AR(1) process). This yielded a continuous time series that preserved both the spatial structure across sensors and the natural smoothness of the signal over time. For the classifier patterns, same as in the example code, we first created a common pattern (i.e. akin to a visual processing ERP) by drawing from a normal distribution for each sensor. Then we created 10 stimulus-specific patterns by drawing from a normal distribution once again and adding to the common pattern for each class. Then we simulated eighteen noisy trials per class, by once again adding noise to the components for each trial with a ratio of 4:1, on which we trained a classifier. Additionally, half of the patterns had additional normal noise added to them. Null data was created by sampling from a normal distribution and adding with a ratio of 1:1. We used a logistic regression with default settings and L1 regularization (C=1). To insert the events into the synthetic resting state, we used the same procedure as in our main simulation. We inserted the stimulus-specific patterns in varying densities to the data, with each event simulating one transition from the graph. Density was again inversely scaled by sampling from the real performances of our participants. Diverging from the original MATLAB code, we did not add a jitter to the time lag and only simulated up to a density of 200 min$^{-1}$, as the density of 2000 min$^{-1}$ used in the original code necessarily introduces a lot of overlap of replay events. Finally, as in the previous synthetic simulation, we induced a temporal autocorrelation in the predicted resting state probabilities by adding a noise vector with a temporal dependency of 0.95. Note, as there was no inter-individual difference in magnitude of sequenceness scores (as they were simulated at the same scale), we did not apply z-scoring to the sequenceness values (in line with the original simulation).

To assess discriminability of the inserted patterns, we extracted classifier probabilities at time points at which we knew a true pattern was present. This was naturally the case during the peak decodability of the localizer and at time points during the simulations in which we had inserted a pattern. For the localizer, we used cross-validation to extract probability estimates for the target class and all other classes of left-out trials. These can be seen as an empirical upper bound of how well patterns are discriminable, as we hypothesize that patterns during a reactivation event will be less pronounced than during visual processing. To assess how far apart the probability distributions of target classes and other classes are, we used the Wasserstein distance (*Ramdas et al., 2017*; *Villani, 2009*). The Wasserstein distance measures how much 'work' would be necessary to transform a distribution into another, also called 'earth-movers-metric'. A larger Wasserstein distance $W$ should reflect a better pattern discriminability of the classifier, with 0 denoting no discriminability and higher values reflecting more separate distributions. However, note this approach only yields a rough estimate and has the caveat that pattern separability is effectively acting on a trial basis (i.e. a trial with low probabilities could still be separable even if the winner class is only slightly higher), while we compare probability distributions across all trials/timepoints. We use it nevertheless in favor of, for example, taking the ratio within a trial as it allows for a scale-invariant approach of comparing distributions of unequal

sample size. Finally, to simulate increased pattern strength, we simply multiplied the inserted patterns by a specific scalar and assessed the Wasserstein distance of probabilities after insertion.

## Acknowledgements

This research was supported by an Emmy-Noether research grant by the DFG (FE1617/2-1), a Consolidator Grant awarded by the ERC (https://doi.org/10.3030/101170886), both awarded to GBF and a project grant by the DGSM as well as a doctoral scholarship of the German Academic Scholarship Foundation, both awarded to SK. Additionally, we want to thank the ZIPP core facility of the Central Institute of Mental Health for their generous support of the study. Finally, we want to thank Prof. T Behrens, Dr. Z Kurth-Nelson and Dr. M Sablé-Meyer, as well as A Wong for the fruitful discussions regarding the novel sign-flip permutation test.

## Additional information

### Funding

| Funder | Grant reference number | Author |
|---|---|---|
| Deutsche Forschungsgemeinschaft | 10.3030/101170886 | Gordon B Feld |
| Deutsche Forschungsgemeinschaft | FE1617/2-1 | Gordon B Feld |
| Deutsche Gesellschaft für Schlafmedizin | | Simon Kern |
| German National Academic Foundation | | Simon Kern |

The funders had no role in study design, data collection and interpretation, or the decision to submit the work for publication.

### Author contributions

Simon Kern, Conceptualization, Data curation, Software, Formal analysis, Investigation, Visualization, Methodology, Writing – original draft, Project administration, Writing – review and editing; Juliane Nagel, Data curation, Methodology; Lennart Wittkuhn, Data curation, Methodology, Writing – review and editing; Steffen Gais, Conceptualization; Raymond J Dolan, Conceptualization, Supervision, Writing – review and editing; Gordon B Feld, Conceptualization, Resources, Supervision, Funding acquisition, Project administration, Writing – review and editing

### Author ORCIDs

Simon Kern ⓘ https://orcid.org/0000-0002-9050-9040
Lennart Wittkuhn ⓘ https://orcid.org/0000-0003-2966-6888
Raymond J Dolan ⓘ https://orcid.org/0000-0001-9356-761X
Gordon B Feld ⓘ https://orcid.org/0000-0002-1238-9493

### Ethics

Informed consent, consent to publish and share anonymized data was obtained. Ethical approval was obtained from the Ethics commitee II at the University of Heidelberg (2020-609-AF 5).

Reviewer #1 (Public review): https://doi.org/10.7554/eLife.108023.3.sa1
Reviewer #2 (Public review): https://doi.org/10.7554/eLife.108023.3.sa2
Reviewer #3 (Public review): https://doi.org/10.7554/eLife.108023.3.sa3
Author response https://doi.org/10.7554/eLife.108023.3.sa4

## Additional files

### Supplementary files
MDAR checklist

### Data availability
Data availability MaxFiltered and anonymized MEG raw data are available at Zenodo in two parts: Localizer data is available as v1 of https://doi.org/10.5281/zenodo.8001755. Resting state data is available as v2 of https://doi.org/10.5281/zenodo.15629081. Code availability The code of the analysis as well as the experiment paradigm and the stimulus material are available at https://github.com/CIMH-Clinical-Psychology/DeSMRRest-TDLM-Simulation, copy archived at *Kern, 2026*.

The following previously published datasets were used:

| Author(s) | Year | Dataset title | Dataset URL | Database and Identifier |
|---|---|---|---|---|
| Kern S | 2025 | Challenges in Replay Detection by TDLM in Post-Encoding Resting State | https://doi.org/10.5281/zenodo.15629081 | Zenodo, 10.5281/zenodo.15629081 |
| Kern S | 2023 | Reactivation strength during cued recall is modulated by graph distance within cognitive maps | https://doi.org/10.5281/zenodo.8001755 | Zenodo, 10.5281/zenodo.8001755 |

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

# Appendix 1

## Replay characteristic and TDLM behavior

In this section, we clarify the behavior of TDLM under three specific conditions. We utilize simulations to demonstrate that the TDLM sequenceness metric is primarily a function of the total number of transitions detected within an analysis window.

### Bursty vs uniformly distributed replay: invariance to temporal distribution of replay

Previous research has indicated that replay occurs in bursts and that the temporal distribution within the resting state might not be uniform. Here, we simulated replay either with uniform temporal distribution for each replay event or a more 'bursty' distribution, which focuses replay in specific areas of the recording. The same number of transitions has been inserted in both cases (taking into account the refractory period between individual replay events). As can be seen in *Appendix 1—figure 1*, TDLM is blind to temporal distribution of transitions within the analysis window. NB this only holds true if no overlap between events occurs, that is the refractory period is regarded. If there is an overlap of different replay events, for example during a burst, a reduction in sensitivity is expected. Additionally, restricting the analysis window to a short period at which a burst is to be expected would increase the sensitivity as the time window decreases.

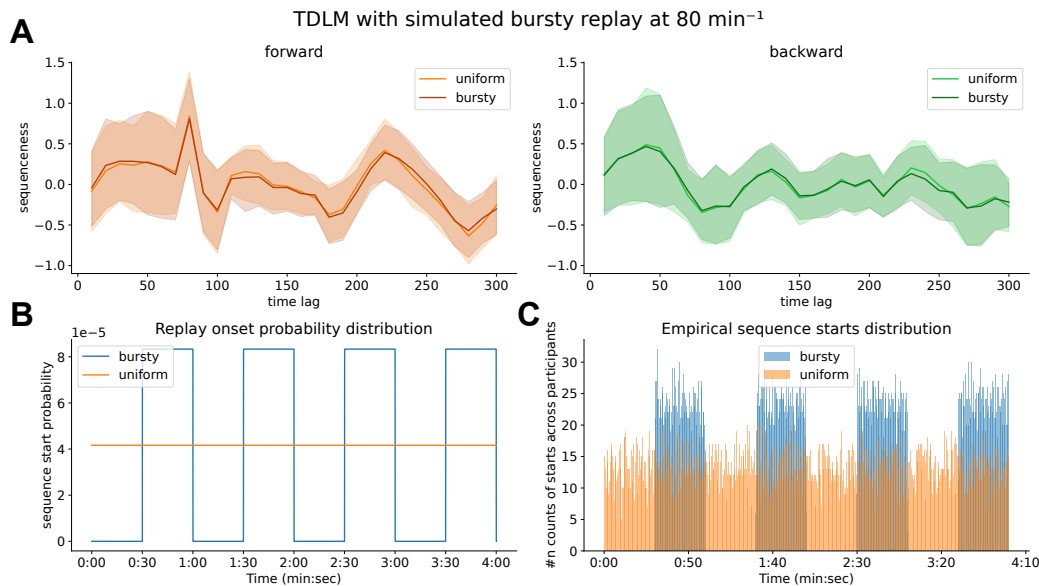

**Appendix 1—figure 1.** TDLM is invariant to the macro-distribution of sequences within the recording. (**A**) Forward and backward sequenceness of replay events that were uniformly distributed (light orange/green) or that were 'bursty', that is temporally constrained to specific areas (dark orange/green). Shaded areas denote the standard error across participants. (**B**) Probability distribution from which the starting points for each replay event were drawn. (**C**) Histogram of empirical replay starting positions across all participants.

### Tripled replay vs doublet replay: invariance towards length of replay

We intentionally designed our study to encourage replay of triplets. However, this begs the question as to whether it matters if triplets or individual chunks of a sequence are replayed at different time points? Here, we simulated two scenarios. In one, we inserted a replay of single transitions alone with a refractory period, for example A->B and separate B->C transitions. In a second scenario, we simulate the replay of chained triplets, for example A->B->C, with a distance of 80 ms each. Importantly, we kept the number of transitions constant, that is A->B, … B->C and where A->B->C would both have two transitions. This creates a context wherein a 4-min resting state would have

~100 events of A->B->C inserted and ~200 events of A->B or B->C, such that in both cases this results in the same number of single step transitions. We found both are equivalent, with TDLM agnostic to the length of sequence trains, that is it does not matter if replay is chunked or chained under the assumption that the number of transitions remains fixed, as can be seen in *Appendix 1— figure 2*.

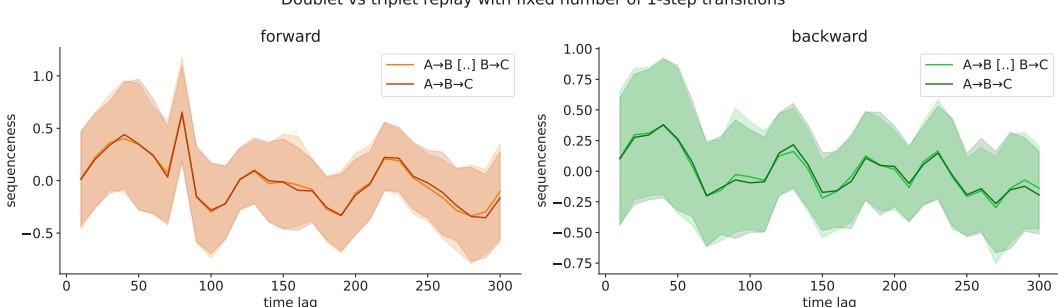

**Appendix 1—figure 2.** TDLM is invariant to the length of sequence replay trains under an assumption that the number of target transitions (e.g. single steps) is fixed. We simulated replay either as two temporally separate A->B, B->C events (light orange/green) or as a single A->B->C event (dark orange/green), both yielding equivalent sequenceness. Shaded areas denote the standard error across participants.

## Full vs actual remembered transitions: invariance to which specific states are replayed

From literature, we know that replay is increased after learning and that less stable memories are replayed more often. We simulated this effect by scaling our replay density inversely with performance. However, for simplicity, in our simulation, we inserted sampled transitions from all valid transitions given by the graph structure, that is the following transitions were valid: However, this meant that some participants would have transitions inserted that they didn't actually remember. To show that this would not change results, we simulated two scenarios: In the full sequence scenario, all valid graph transitions are inserted (i.e. all participant's replay is sampled from 'A->B, B->C, C->D, D->E, E->F, F->G, G->E, E->H, H->I, I->B, B->J, J->A'). In the second scenario (memorized transitions), we only replayed transitions that the participant actually retrieved correctly during the post-resting state testing sessions (i.e. a participant's replay would have been sampled from 'A->B, B->C, G->E, E->H, H->I', if those were the ones he remembered). In both scenarios, the number of events is kept constant. The results are equivalent, as can be seen in *Appendix 1—figure 3*. NB this only holds under the assumptions that classifiers are equally good at decoding each class.

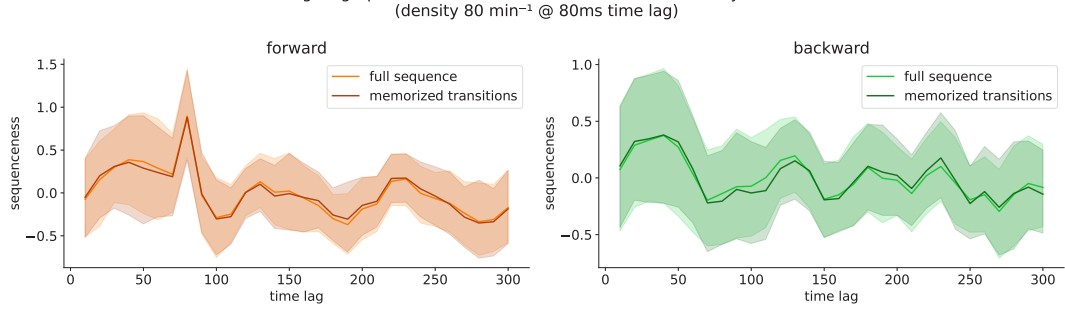

**Appendix 1—figure 3.** TDLM is insensitive towards which transitions are replayed and only sensitive to how many transitions are detected in total. Here we simulate transitions either sampled from the full graph (light orange/green) or participant-specific transitions of trials that participants correctly remembered (dark orange/green). Shaded areas denote the standard error across participants.

## Simulating multi-step replay: density increases while number of transitions stays similar

From rodent literature, we know that a replay event often consists of several reactivation events. However, in our study, to keep the simulation as basic as possible and to keep consistent with previous simulations, we intentionally only simulated single transitions. As shown above, TDLM is invariant to the sequence length that is replayed and only sensitive to the number of transitions. Our definition of density scales inversely to the number of transitions in a replay event, giving the impression of an increased sensitivity. Here, we simulate how TDLM behaves under multi-step replay events. We simulated replay with increasing number of transitions while still preventing overlap by expanding the refractory period accordingly. As the number of transitions per replay events scales with the step size, the replay density to find replay is linearly decreasing as well (see *Appendix 1— figure 4*). Nevertheless, TDLM's sensitivity is primarily governed by the total number of target transitions present in the data, with all three cases becoming significant at around 70–80 single-step transitions found. Additionally, multi-step sequences pose another problem: If we assume that a classifier has a decoding accuracy of 35%, successfully detecting two events in a row would have an upper bound chance of ~12% and decreases exponentially for longer step size. As shown above, our simulation likely overestimates pattern discriminability and therefore gives a biased view of how multi-step sequences would be detected by TDLM.

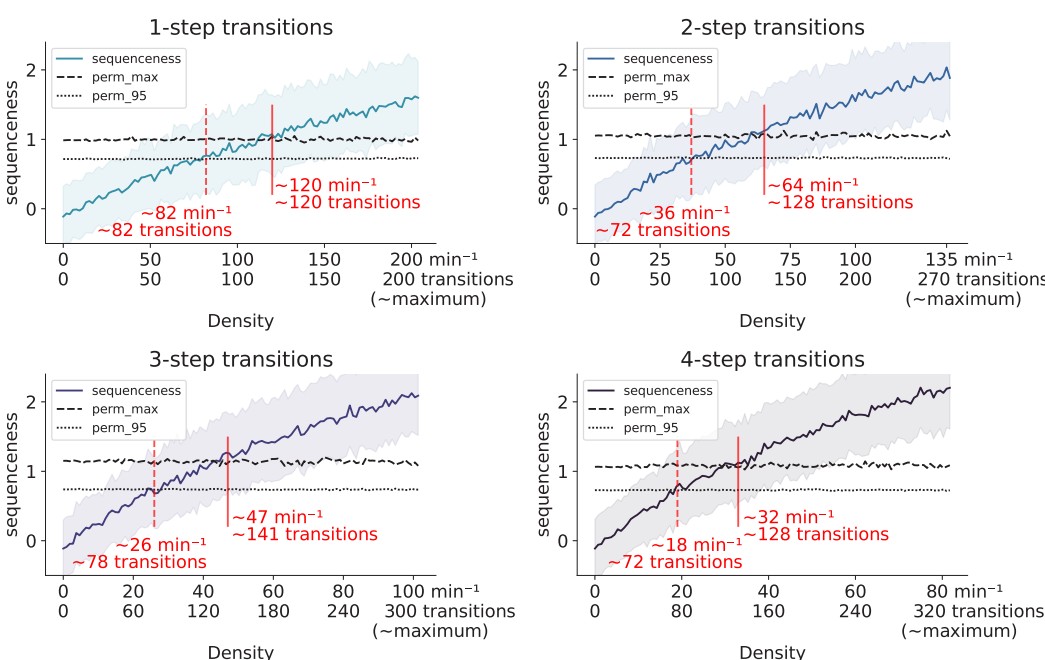

**Appendix 1—figure 4.** Sequenceness for longer multi-step replay. Here, we inserted sequence trains of longer replay (e.g. A->B->C->D->E for the 4-step case) without keeping the number of transitions fixed. To prevent overlap, an increase in length of replay is accompanied by a longer refractory window, which leads to a reduced number of maximum replay events per minute. TDLM becomes more sensitive under the assumption that (i) multiple sequence steps are reactivated and (ii) all sequence steps are decoded accurately. Notably, as mentioned before, only the number of single-step transitions is relevant for TDLM's sensitivity. The indicated line is sequenceness at the simulated 80 ms time lag across different time lags, with participants' standard error in the shaded area.

