## [Editor Report · eLife Assessment]

This study presents **valuable** findings on the ability of a state-of-the-art method, Temporally Delayed Linear Modelling (TDLM), to detect the replay of sequences in human memory. The investigation provides **compelling** evidence that TDLM has significant limitations in its sensitivity to detect replay in extended (minutes-long) rest periods. The work will be of strong interest to researchers investigating memory reactivation in humans, especially using iEEG, MEG, and EEG.

---

## [Referee Report · Reviewer #1 (Public review)]

Summary:

Participants learned a graph-based representation, but, contrary to the hypotheses, failed to show neural replay shortly after. This prompted a critical inquiry into temporally delayed linear modeling (TDLM)--the algorithm used to find replay. First, it was found that TDLM detects replay only at implausible numbers of replay events per second. Second, it detects replay-to-cognition correlations only at implausible densities. Third, there are concerning baseline shifts in sequenceness across participants. Fourth, spurious sequences arise in control conditions without a ground truth signal. Fifth, the revised manuscript adapts a previously published synthetic simulation to show that previous validations/support of TDLM may have overestimated TDLM sensitivity because synthetic assumptions can produce unrealistically high pattern separability and reduced baseline confounds.

Strengths:

- This work is meticulous and meets a high standard of transparency and open science, with preregistration, code and data sharing, external resources such as a GUI with the task and material for the public.

- The writing is clear, balanced, and matter-of-fact.

- By injecting visually evoked empirical data into the simulation, many surface-level problems are avoided, such as biological plausibility and questions of signal-to-noise ratio.

- The investigation of sequenceness-to-cognition correlations is an especially useful add-on because much of the previous work uses this to make key claims about replay as a mechanism.

- In the revised version, the authors foreshadow ways to improve sequenceness detection by introducing a sign-flipping analysis.

Weaknesses:

Many of the weaknesses are not so much flaws in the analyses, but shortcomings when it comes to interpretation and a lack of making these findings as useful as they could be. Furthermore, as I will explain below, some weaknesses have been partially improved in the last round of revisions.

- I found the bigger picture analysis to be lacking, though improved in the latest version. Let us take stock: in other work during active cognition, including at least one study from the Authors, TDLM shows significant sequenceness. But the evidence provided here suggests that even very strong localizer patterns injected into the data cannot be detected as replay except at implausible speeds. How can both of these things be true? Assuming these analyses are cogent, do these findings not imply something more destructive about all studies that found positive results with TDLM? In the revisions, the manuscript concentrates a bit more on criteria that influence detection of sequences, though it is still not entirely clear what consequences there are for previous work.

- All things considered, TDLM seems like a fairly vanilla and low assumption algorithm for finding event sequences. Although the authors have improved their discussion of "boundary conditions" or factors for why TDLM might fail, it remains not fully clear to what extent the core problem is TDLM on an algorithmic/mathematical level (intrinsic factor), vs data quality, power, window size (extrinsic factors).

- The new sign-flip analysis underscores the authors' goal of being solution-oriented, though it is important to emphasize that a comprehensive way forward is not yet provided. This is fine, but the manuscript could be improved further through a concrete alternative or a revised version of the original approach.

---

## [Referee Report · Reviewer #2 (Public review)]

Summary:

Kern et al. investigated whether temporally delayed linear modeling (TDLM) can uncover sequential memory replay from a graph-learning task in human MEG during an 8 minute post-learning rest period. After failing to detect replay events, they conduct a simulation study in which they insert synthetic replay events, derived from each participants' localizer data, into a control rest period prior to learning. The simulations suggest that TDLM only reveals sequences when replay occurs at very high densities (> 80 per minute) and that individual differences in baseline sequenceness may lead to spurious and/or lacklustre correlations between replay strength and behavior.

Strengths:

The approach is extremely well documented and rigorous. The authors have done an excellent job re-creating the TDLM methodology that is most commonly used, reporting the different approaches and parameters that they used, and reporting their preregistrations. The hybrid simulation study is creative and provides a new way to assess the efficacy of replay decoding methods, and its comparison to earlier published TDLM simulations is particularly useful. The authors remain measured in the scope/applicability of their conclusions, constructive in their discussion, and end with a useful set of recommendations for how to best apply TDLM in future studies. I also want to commend this work for not only presenting a null result, but thoroughly exploring the conditions under which such a null result is expected. I think this paper is interesting and will be generally quite useful for the field.

In the revised version, the authors have adequately addressed each of the weaknesses I raised previously. In brief, they:

(i) Added new power analyses of sequenceness for bootstrapped sample sizes, along with a new permutation test (Supplemental Fig 11),

(ii) Qualified their conclusions with added limitations and clarified several points that I found previously unclear,

(iii) Added several new analyses to the Appendices

(iv) Demonstrated that previous simulations validating TDLM overestimated TDLM sensitivity relative to the hybrid simulation.

(v) Added a new and extensive appendix on the relationship between TDLM and replay characteristics.

Weaknesses:

The remaining weaknesses of the work relate primarily to explaining the cause of measured non-random fluctuations in the simulated correlations between replay detection and performance at different time lags, as well as a lack of general recommendations of parameter choices for applying TDLM in future work. But these are minor weaknesses that can be left to future work.

---

## [Referee Report · Reviewer #3 (Public review)]

Summary:

Kern et al. critically assess the sensitivity of temporally delayed linear modelling (TDLM), a relatively new method used to detect memory replay in humans via MEG. While TDLM has recently gained traction and been used to report many exciting links between replay and behavior in humans, Kern et al. were unable to detect replay during a post-learning rest period. To determine whether this null result reflected an actual absence of replay or sensitivity of the method, the authors ran a simulation: synthetic replay events were inserted into a control dataset, and TDLM was used to decode them, varying both replay density and its correlation with behavior. The results revealed that TDLM could only reliably detect replay at unrealistically (not-physiological) high replay densities, and the authors were unable to induce strong behavior correlations. These findings highlight important limitations of TDLM, particularly for detecting replay over extended, minutes long time periods.

Strengths:

Overall, I think this is an extremely important paper, given the growing use of TDLM to report exciting relationships between replay and behavior in humans. I found the text clear, the results compelling, and the critique of TDLM quite fair: it is not that this method can never be applied, but just that it has limits in its sensitivity to detect replay during minutes long periods. Further, I greatly appreciated the authors efforts to describe ways to improve TDLM: developing better decoders and applying them to smaller time windows.

The power of this paper comes from the simulation whereby the authors inserted replay events and attempted to detect them using TDLM. Regarding their first study, there are many alternative explanations or possible analysis strategies that the authors do not discuss; however, none of these are relevant if replayed, under conditions where it is synthetically inserted, cannot be detected.

Further, the authors provide a simulation and series of analyses aimed at replicating previous TDLM-based replay studies. They demonstrate methodological flaws, and show that previous simulations greatly overestimated the sensitivity of TDLM. This work emphasizes the need to cast a critical eye over both past and future studies applying TDLM to detect replay.

Finally, the authors are relatively clear about which parameters they chose, why they chose them, and how well they match previous literature (they seem well matched); and provide suggestions for how others can determine the best parameters for TDLM within their own experimental contexts.

Comments on revisions:

The authors thoroughly addressed my previous comments; the added analyses and discussion significantly strengthen the paper's clarity, utility, and impact.

---

## [Author Response]

The following is the authors’ response to the original reviews.

**Public Reviews:**

**Reviewer #1 (Public review):**
(1) I found the bigger picture analysis to be lacking. Let us take stock: in other work, during active cognition, including at least one study from the Authors, TDLM shows significance sequenceness. But the evidence provided here suggests that even very strong localizer patterns injected into the data cannot be detected as replay except at implausible speeds. How can both of these things be true? Assuming these analyses are cogent, do these findings not imply something more destructive about all studies that found positive results with TDLM?

Our focus here is on advancing methodology. Given the diversity of tasks and cognitive states in the TDLM literature, replay could exceed detection thresholds under specific conditions—especially when true event durations align with short analysis windows. While a comprehensive re-analysis of prior datasets is beyond our scope, we agree a concise synthesis can strengthen the paper.

The previous TDLM literature uses a diverse set of tasks and addresses a broad spectrum of cognitive constructs/processes. As we acknowledge, it is perfectly possible that replay bursts in short time windows are well detectable by TDLM. However, we acknowledge that some commentary on this is warranted and have added the following paragraph to the discussion that addresses “improving TDLMs sensitivity”:

“Finally, what do our simulations imply for the broader MEG replay literature? Our implementation successfully detects replay when boundary conditions are met, as shown in the simulation. But sensitivity depends critically on high fidelity between the analysis window and the density of replay events. A systematic evaluation of these conditions as they apply to prior studies remains beyond the scope of the current paper. Instead, our focus is on delineating boundary conditions that we hope will motivate conduct of power analyses in future work as well as inclusion of simulations that approximate realistic experimental conditions.”

(2) All things considered, TDLM seems like a fairly 'vanilla' and low-assumption algorithm for finding event sequences. It is hard to see intuitively what the breaking factor might be; why do the authors think ground truth patterns cannot be detected by this GLM-based framework at reasonable densities?

We agree with the overall sentiment of the referee. Our intuition is that one of the principal shortcomings of the method relates to spurious sequenceness induced by unknown factors at baseline, and poor transfer of the decoder to other modalities. and have a rough understanding of how they occur, we are currently not in a position to identify their nature. Note that we believe that these confounders are not exclusive to TDLM but are potentially threatening to all kinds of sequenceness analysis of longer time series that rely on decoders. Indeed, we suspect that classifier training is another bottleneck, as we don’t know the exact nature of the representations that are replayed, including the degree of overlap there is with a commonly used visual localizer. That said, this is not of relevance for the simulation in so far as we insert patterns that exceed the pattern strength in the localizer.

Finally, a potential major drawback is the permutation test for significance testing. As the original authors of TDLM have noted, the current test which permutes states is overly conservative. It measures fixed effects and as it only considers the group level mean it is accordingly easily biased by individual outliers. This we have tried to account for by z-scoring sequenceness scores. We have also conferred on this with some of the authors of TDLM and discussed a yet unpublished method that aims to address this exact issue. The proposed new method uses a sign-flip permutation test at a group level and therefore implements a random-effects model of the data. This significance test has markedly increased power while still controlling for FWER. However, while we show in our power analysis that the new method is indeed more sensitive, it does not materially change the interpretation of the data. We have included this novel method in the paper and added it into the main analysis and most of the simulations.

(3) Can the authors sketch any directions for alternative methods? It seems we need an algorithm that outperforms TDLM, but not many clues or speculations are given as to what that might look like. Relatedly, no technical or "internal" critique is provided. What is it about TDLM that causes it to be so weak?

We believe there are several shortcomings and bottlenecks within TDLM that need to be evaluated and improved. While we highlight these issues in the discussion section titled “Improving TDLMs sensitivity,” we agree that we should provide a clearer outline of its current shortcomings. We have now added to the discussion to expand on that we think needs improvement (‘fixed time lag’) and also add a summary statement at the end of the relevant paragraph to recap the main issues needed for an improved successor method. The new paragraphs read:

“Lastly, there are certain assumptions that TDLM makes that might not hold (see Methods Study II): Current implementations look for a fixed time lag that is the same across all participants and between all reactivation events. If time lags differ across participants, TDLM will fail to find them. Similarly, TDLM assumes a fixed sequence order and is not robust against slight within-sequence permutations or in-sequencemissing reactivation events. However, from other data sources., such as hippocampal place cell recordings, it is known that such permutations can occur where some states are skipped or fail to decode during replay. Similarly, it is assumed that each reactivation event lasts between 10-30 milliseconds, but the true temporal evolution of reactivation measured by TDLM is currently unknown. Future method development might focus on improving invariance to these assumptions.

[…]

In summary, there are several areas where TDLM might be improved, including a restriction in its search space, improvement in classifiers, a validation of localizer representation transfer to other domains (e.g. memory representations), and the extension of TDLM to render it more robust against violations of its core assumptions.”

**Reviewer #2 (Public review):**
Weaknesses:The sample size is small (n=21, after exclusions), even for TDLM studies (which typically have somewhere between 25-40 participants). The authors address this somewhat through a power analysis of the relationship between replay and behavioural performance in their simulations, but this is very dependent on the assumptions of the simulation. Further, according to their own power analysis, the replay-behaviour correlations are seriously underpowered (~10% power according to Figure 7C), and so if this is to be taken at face value, their own null findings on this point (Figure 3C) could therefore just reflect under sampling as opposed to methodological failure. I think this point needs to be made more clearly earlier in the manuscript.

We agree with the referee that our sample is smaller than previous studies due to participant exclusion criteria. However, the take-away message from our behavioural simulation and bootstrapping is that even with larger sample sizes, it is difficult to overcome baseline fluctuations of sequenceness, even if very strong replay patterns were detectable and sample sizes were of similar size to that of previous studies. Therefore, we are not convinced that that our null findings are fully explained by the smaller sample size compared to that of previous studies, Additionally, we show that even within the range of other studies, similar power would have been expected (Supplement Figure 11). However, it is true that in general null findings can be explained by under-sampling, under the assumption that an effect is present. To amplify this point, we have added the following to the Figure 3C:

“[…]. NB, however, as our simulation shows, correlations of sequenceness with behavioural markers are likely to be underpowered and occur only with very high replay rates or much higher sample size. See our simulation discussion for a more detailed explanation on how correlations may be inherently biased, where fluctuations in baseline sequenceness overshadow individual scaling with behavioural markers.”

Furthermore, we have added the following paragraph to the discussion to highlight this point and refer to a power analysis we have now added to the supplement (see next answer):

“Sample sizes in previous TDLM literature usually range between 20 to 40 participants. A bootstrap power analysis shows that even at those sample sizes, power would remain low unless unrealistically high replay rates are assumed (Supplement Figure 11). Our bootstrap simulation shows that a correlation analysis between sequenceness and behaviour would in these cases be drastically underpowered, even under an assumption of high replay densities.”

Finally, we have added a remark about the sample size to the limitations section, as naturally, an increase in sample size would yield higher power:

“Finally, while initially planning for thirty participants, due to exclusion criteria, our study featured fewer participants than most previous studies using TDLM (i.e. usually 25-40, but 21 in our study). While we are confident that our simulation results hold under these sample sizes, as sample sizes of other studies show comparable power to ours (Fehler! Verweisquelle konnte nicht gefunden werden.), we cannot fully rule out a possibility that our null-findings are explained by a lack in power alone.”

Relatedly, it would be very useful if one of the recommendations that come out of the simulations in this paper was a power analysis for detecting sequenceness in general, as I suspect that the small sample size impacts this as well, given that sequenceness effects reported in other work are often small with larger sample sizes. Further, I believe that the authors' simulations of basic sequenceness effects would themselves still suffer from having a small number of subjects, thereby impacting statistical power. Perhaps the authors can perform a similar sort of bootstrapping analysis as they perform for the correlation between replay and performance, but over sequenceness itself?

We agree with the referee that this, in principle, is a great idea. However, the way that significance thresholds are calculated poses a conceptual problem for such an analysis: as for significance threshold we are defining the maximum sequenceness value across all participants, all time lags and all permutations. This sequenceness value is compared against the mean of all participants, disregarding the standard deviation. This maximum threshold would not change if we bootstrapped some of our samples. Additionally, the 95% would also not change significantly. To illustrate this point, we have added this analysis to the supplement, as Supplement Figure 10. However, the new sign-flip permutation test we now include allows for such a comparison, as it takes variance between participants into account as well! We have included all three variants of the power analysis and the figure description now reads:

“Supplement Figure 11 Power analysis of sequenceness significance for bootstrapped samples sizes. (A) Powermap for state-permutation thresholds. However, here the bootstrap approach suffers from a conceptual problem: significance thresholds are defined by the permutation maximum and/or 95-percentile of the maximums across all sequence-permutations across participants. If we resample bootstrap-participants from our existing pool, the maximum thresholds computed will remain relatively stable across resampled participants, as it only compares against the mean and disregards the standard deviation. (B) The newly presented statistical approach is significantly more sensitive at higher sample sizes. Note that even then, 80% power is only reached with replay density of higher than 50 min-1 at a sample size of 60 participants. Additionally, the sign-flip permutation test assumes that the mean is at zero. As we observed a non-zero mean due to spurious oscillations, we subtracted the mean sequenceness of the baseline condition from each participant before permuting to achieve a null distribution with mean zero, as otherwise, we would have found significant replay effects in the baseline condition at increasing sample size. Nevertheless, due to the higher sensitivity, the new sign-flip test is recommended over the previous sequence-permutation-based test. Colours indicate the power from 0 to 1 for different bootstrapped sample sizes and densities. 80% power thresholds are outlined in black.”

The task paradigm may introduce issues in detecting replay that are separate from TDLM. First, the localizer task involves a match/mismatch judgment and a button press during the stimulus presentation, which could add noise to classifier training separate from the semantic/visual processing of the stimulus. This localizer is similar to others that have been used in TDLM studies, but notably in other studies (e.g., Liu, Mattar et al., 2021), the stimulus is presented prior to the match/mismatch judgment. A discussion of variations in different localizers and what seems to work best for decoding would be useful to include in the recommendations section of the discussion.

We agree and thank the referee for raising this issue. Note, we acknowledge we forgot to mention that these trials were excluded from classifier training. Our rationale of presenting the oddball during stimulus presentation, and not thereafter, was an assumption that by first presenting the audio and then the visual cue we would create more generalized representations that would be less modalitydependent. However, importantly, we excluded all trials that were oddballs from localizer training. Therefore we assume that this particular design choice will not greatly affect the decoder training. If some motor-preparation activity is present during the stimulus presentation, then it should be present equally across all trials and hence be ignored by the classifier as we balanced the transitions between images. We now added this information to the main text:

“In each trial, a word describing the stimulus was played auditorily, after which the corresponding stimulus was shown. In ~11% of cases, there was a mismatch between word and image (oddball trials), and these trials were excluded from the localizer training.” Additionally in the methods section: “These oddball-trials were excluded from all further analysis and decoder training.”

Nevertheless, we agree that the extant variety in localizer designs is underdiscussed where many assumptions of classifier training are not, as yet, fully validated. We have added a sentence highlighting different oddball paradigms to the section on the discussion of localizers and also add a summary statement with recommendations. The passage now reads:

“Additionally, a wide variety of oddballs has been used (e.g. upside-down, scrambled, or mismatched images, cues presented visually, as words, auditorily, etc), and at this time it is unclear if these affect the representations that the classifier learns [...] In summary, we would expect a multimodal categorical localizer, and a classifier that isn’t trained on a specific timepoint, to generalize best.”

Second, and more seriously, I believe that the task design for training participants about the expected sequences may complicate sequence decoding. Specifically, this is because two images (a "tuple") are shown together and used for prediction, which may encourage participants to develop a single bound representation of the tuple that then predicts a third image (AB -> C rather than A -> B, B -> C). This would obviously make it difficult to (i) use a classifier trained on individual images to detect sequences and (ii) find evidence for the intended transition matrix using TDLM. Can the authors rule out this possibility?

We thank the reviewer for raising a possibility we have not considered! While there is some evidence that a single bound representation would have overlap with its constituents (especially before long term-consolidation) and therefore be detectable by the classifiers, we acknowledge the possibility that individual classifiers would fail to be sensitive to such a compound representation. In fact we find in the retrieval data some evidence for a combined replay of representations (where representations are replayed seemingly at the same time, see Kern 2024). We have added such a possibility to the interims-discussion of Study 1 as a qualification . However, this does not change the results or interpretation of our simulation which we consider is a key message of the paper.

The relevant segment in the discussion section now reads:

“Additionally, given that the stimuli were presented in combined triplets, participants may have formed a singular representation of associated items and subsequently replayed these (e.g., AB→C), instead of replaying item-by-item transitions (A→B→C). Under such a scenario, a classifier trained on individual items may fail to detect these newly formed bound representations, particularly if they diverge strongly from the single-item patterns. In our previous study where we address retrieval (Kern et al., 2024) we found that states were to varying extent co-reactivated, yet classifiers trained on single items retained sensitivity to detect these combined reactivation events. Consistent with this, prior work suggests that unified representations retain overlap with their constituent item representations (Dennis et al., 2024; Liang et al., 2020), however, there’s also evidence that different brain regions are involved if representational unitization occurs (Staresina & Davachi, 2010), potentially confusing classifiers. Therefore, we cannot exclude that rest-related consolidation replays engendered unitized representations that were insufficiently captured by our singleitem classifiers.“

Participants only modestly improved (from 76-82% accuracy) following the rest period (which the authors refer to as a consolidation period). If the authors assume that replay leads to improved performance, then this suggests there is little reason to see much taskrelated replay during rest in the first place. This limitation is touched on (lines 228-229), but I think it makes the lack of replay finding here less surprising. However, note that in the supplement, it is shown that the amount of forward sequenceness is marginally related to the performance difference between the last block of training and retrieval, and this is the effect I would probably predict would be most likely to appear. Obviously, my sample size concerns still hold, and this is not a significant effect based on the null hypothesis testing framework the authors employ, but I think this set of results should at least be reported in the main text.

We disagree that an absence or presence of replay might be inferred from an absolute memory enhancement. While consolidation can lead to absolute improvement of performance in, for example, motor memory domains one formulation is that in declarative learning tasks replay stabilizes latent memory traces, and in such a scenario would not necessarily lead to a boosted performance. While many declarative consolidation studies report an increase of performance compared to a control condition (i.e. without a consolidation window), this does not necessarily entail an absolute performance increase, as replay might just act to protect against loss of memory traces. Therefore, the modest increase we observe does not inference as to the presence of absence of replay absent a proper control condition.

We did expect to find a correlation between replay and individual behavioural. Indeed, a weak correlation with performance and sequenceness can be detected. However, as we also show any such correlation is overshadowed by baseline fluctuations in sequenceness such that its overall validity is questionable, even under very high replay rates. We are therefore circumspect about this correlation, even if it was significant. Therefore, in the discussion, we chose to refrain from putting much focus on this correlation. Nevertheless, we do add a short statement to the corresponding figure label, discussing this precise issue. The segment now reads:

“While we found a non-significant relation between a memory performance enhancement and post-learning forward sequenceness we are cautious not to overinterpret these results. As in the section “Correlation with behaviour only present at high replay speeds” the noted correlational measure oscillates heavily with baseline sequenceness fluctuations, and any true replay effect is likely to be overshadowed by such fluctuations.”

I was also wondering whether the authors could clarify how the criterion over six blocks was 80% but then the performance baseline they use from the last block is 76%? Is it just that participants must reach 80% within the six blocks *at some point* during training, but that they could dip below that again later?

We thank the reviewer for highlighting this point: The first block wherein participants reached >80% ended the learning blocks. After a maximum of six blocks the learning session was ended regardless of performance. Therefore, some participants’ learning blocks were ended after six blocks and without them reaching a performance of 80%.. While we described this in the Methods section, it was missing from the Results Study I section, which now contains:

“[...] Participants then learned triplets of associated items according to a graph structure. Within the learning session, participants performed a maximum of six learning blocks, but the session was stopped if participants reached 80% memory performance criterion learning,, up to a memory performance criterion of 80% (see Methods for details)”

The Figure 2 description now contains

“[...] Participants’ completed up to six blocks of learning trials. After reaching 80% in any block, no more learning blocks were performed (criterion learning) [...]”

Lastly, there was a mistake in the Behavioural results section, which stated “All thirty participants, except one, [..] to criterion of 80%.” This is an error. In our preregistration, we defined to only include participants that successfully learned anything at all above chance. Here,we meant that only one participant failed to reach a criterion that we defined as “successful learning”. We fixed it and it now reads

“with an accuracy above 50% (which we preregistered beforehand as an exclusion criterion for “successful learning above chance”).”

Additionally, we have noted this for clarity in the methods section and excuse this mistake:

“Additionally, as successful above-chance learning was necessary for the paradigm, we ensured all remaining participants had a retrieval performance of at least 50% (one participant had to be excluded, but was already excluded due to low decoding performance).”

Because most of the conclusions come from the simulation study, there are a few decisions about the simulations that I would like the authors to expand upon before I can fully support their interpretations. First, the authors use a state-to-state lag of 80ms and do not appear to vary this throughout the simulations - can the authors provide context for this choice? Does varying this lag matter at all for the results (i.e., does the noise structure of the data interact with this lag in any way?)

This was a deliberate choice but we acknowledge the reasoning behind this was not detailed in our initial submission. We chose a lag of 80 millisecond for three reasons: first, it is distant from the 9-11 Hz alpha oscillations we observed in our participants and does not share a harmonic with the alpha rhythm; second, we wanted to get a clear picture of the effect of simulated replay that is as isolated as possible from spurious sequenceness confounders present in the baseline condition. Thus, we chose a lag in which the sequenceness score was close to zero in the baseline condition; thirdly , in this revision, we subtracted the mean sequenceness value of the baseline such that any simulation effects would start, on average, at zero sequenceness. In this way, we could attribute any increase in sequenceness to the experimentally inserted replay, that was independent of spurious oscillations. Finally (but less importantly), as we observed that a correlation of sequenceness with behaviour was fluctuated strongly, for the reason detailed above, we chose a lag in which a correlation was as close as possible to zero. If we had not chosen a lag that adhered to these conditions, we were at risk of measuring simulated replay plus spurious sequenceness confounders.

We have added a sentence to the main text detailing this justification:

“We chose this timepoint (80 msec state to state lag) as its sequenceness value was close to zero in the baseline condition as well as being distant to the observed alpha rhythms of the participants (which varied between ~9-11 Hz). Additionally, we subtracted the mean sequenceness value of the baseline at 80 milliseconds lag such that any simulation effects would, on average, start at zero sequenceness “

Additionally, we now add a more detailed explanation to the methods section.

“This time lag (80 msec) was chosen in order to isolate precisely an effect of the experimentally inserted sequenceness. Thus, we chose a lag at which the mean baseline sequenceness was close to zero and where the correlation with behaviour was low. Additionally, we subtracted the mean sequenceness value (at 80 milliseconds) at baseline from the specific lag recorded for each participant, such that simulation effects would be initialized at zero sequenceness on average enabling any effects to be attributed purely to inserted replay. Additionally, we excluded time lags too close to the alpha rhythms of participants (which varied between ~9-11 Hz) or lags which would have a harmonic with the rhythm.”

Second, it seems that the approach to scaling simulated replays with performance is rather coarse. I think a more sensitive measure would be to scale sequence replays based on the participants' responses to *that* specific sequence rather than altering the frequency of all replays by overall memory performance. I think this would help to deliver on the authors' goal of simulating an "increase of replay for less stable memories" (line 246).

The referee makes an excellent point and our simulations could be rendered more realistic by inserting the actual tuples that participants answered correctly. If we understand the point correctly, there are two different ways replay might be impacted by performance: First, we can conjecture that there is greater replay if memory performance is not saturated. Second, replay only occurs for content that has actually been encoded!

The main reasons why we chose to simulate the entire sequence being replayed for each participant is based on the following. TDLM is implemented such that the amount of replay alone is relevant, and actual transitions are not affecting the results beyond noise. Under the assumption that class-specific classifiers perform equally well, simulating A->B, B->C or simulating A->B, A->B yields equivalent results. However, results can differ if this assumption is violated. By drawing from the entire space of classes we insert, we minimize the risk of some classifiers being worse than others for some participants. For example, if we simulated only A->B for some participant instead of the whole sequence, and by chance classifier A performs suboptimally, we would then introduce additional unwanted variance into our results.

Secondly, from our reading of the literature we infer that replay is increased generally (i.e. density of learning-specific replay is increased) for less stable memories. However, we do not have indicators of memory strength, but only a binary “remembered or not”. As TDLM is invariant to the actual transitions being replayed and only indexes the number of transitions, we chose to ignore which transitions we insert and only scaled the amount of replay.

We have added an analysis to the Appendix that discusses this specific aspect of our study where we show that results are equivalent if we simulate replay of “A->B B->C C->D” or only “A->B A->B A->B A->B”. As we do not know how replay density interacts with memory trace stability, we opted to leave the current simulation as is. The corresponding paragraph and figure description now read:

“From literature we know that replay is increased after learning and that less stable memories are replayed more often. We simulated this effect by scaling our replay density inversely with performance. However, for simplicity, in our simulation, we inserted sampled transitions from all valid transitions given by the graph structure, i.e., the following transitions were valid: However, this meant that some participants would have transitions inserted that they didn’t actually remember. To show that this would not change results, we simulated two scenarios: In the full sequence scenario, all valid graph transitions are inserted (i.e. all participant’s replay is sampled from 'A->B, B->C, C->D, D->E, E->F, F->G, G->E, E->H, H->I, I->B, B->J, J->A'). In the second scenario (memorized transitions) we only replayed transitions that the participant actually retrieved correctly during the post-resting state testing sessions (i.e. a participant’s replay would have been sampled from ‘A->B, B->C, G->E, E->H, H>I’, if those were the ones he remembered). In both scenarios, the number of events is kept constant. The results are equivalent as can be seen in Appendix A Figure 3. NB this only holds under the assumptions that classifiers are equally good at decoding each class.”

[…]

“TDLM is insensitive towards which transitions are replayed and only sensitive to how many transitions are detected in total. Here we simulate transitions either sampled from the full graph (light orange/green) or participant-specific transitions of trials that participants correctly remembered (dark orange/green). Shaded areas denote the standard error across participants.”

On the other hand, I was also wondering whether it is actually necessary to use the real memory performance for each participant in these simulations - couldn't similar goals (with a better/more full sampling of the space of performance) be achieved with simulated memory performance as well, taking only the MEG data from the participant?

The decision to use real memory performance is indeed arbitrary. We could have also used randomly sampled values. However, as we wanted to understand our nullresults better we opted to use real performance to adhere as close as possible to the findings we previously reported. Using uniformly sampled memory performance would be less explanatory w.r.t to our actual results of the resting state data that are reported in the first study we report in the manuscript (Study I).

Nevertheless, our current implementation already presents an approach that samples the entire performance range for the sub-analysis focusing on the correlation with behaviour. Here, in the section on “best-case”-scenario, we implement this such that it spans factors from 1 to 0 (i.e., a participant with 100% performance gets a replay scale factor of 0 and hence no replay simulated, and the worst performing participant with 50% performance has a replay rate multiplied by 1). We scale the amount of replay with this factor. As a correlation is invariant to linear scaling, statistically this is equivalent to stretching the performance distribution from 0 to 100%. We have added a sentence to the methods to provide further focus on this point:

“To assess how performance might affect replay in our specific dataset, we chose to use the original participants’ performance values instead of uniformly sampling the performance space (which ranged from 50 to 100%). However, for the correlation analysis, we additionally added a “best-case” scenario, in which we scale replay from 0 to 1, an approach that is statistically equivalent to scaling values to the full space of possible performance (0 to 100%) (see Results Study II: Simulation).”

Finally, Figure 7D shows that 70ms was used on the y-axis. Why was this the case, or is this a typo?

Thanks, this is indeed a typo, we fixed it.

Because this is a re-analysis of a previous dataset combined with a new simulation study on that data aimed at making recommendations about how to best employ TDLM, I think the usefulness of the paper to the field could be improved in a few places. Specifically, in the discussion/recommendation section, the authors state that "yet unknown confounders" (line 295) lead to non-random fluctuations in the simulated correlations between replay detection and performance at different time lags. Because it is a particularly strong claim that there is the potential to detect sequenceness in the baseline condition where there are no ground-truth sequences, the manuscript could benefit from a more thorough exploration of the cause(s) of this bias in addition to the speculation provided in the current version.

We are currently working on a theoretical basis to explain these spurious sequenceness confounders in the baseline condition. Indeed, in our preliminary work, in certain contexts we can induce significant sequenceness in the absence of any replay signal during baseline. However, this work is at an early stage and we still have some conceptional problems to solve before we are confident enough with these data. We believe at present it would be premature to add these data to the current manuscript. Nevertheless, we now mention these spurious sequenceness confounders to raise awareness for the field and also add greater context to the discussion, highlighting one of the issues that we think is of importance:

“[…] For example, if two classifiers’ probabilities oscillate at 10 Hz but at a different phase, a spurious time lag can be found reflecting this phase shift. We speculate that more complex interactions between classifiers oscillating at different phases are also conceivable.”

In addition, to really provide that a realistic simulation is necessary (one of the primary conclusions of the paper), it would be useful to provide a comparison to a fully synthetic simulation performed on this exact task and transition structure (in addition to the recreation of the original simulation code from the TDLM methods paper).

Thank you for this suggestion! We have now added a synthetic simulation, trying to keep as close as possible to the original simulation code in Liu et al. (2021), while also incorporating our current means of simulating the data (i.e. scaling by performance). We think this synthetic simulation greatly improves the paper and gives weight to our suggestion about the superiority of a hybrid approach. Additionally, it prompted us to look closer at patterns that are inserted in the synthetic simulation and perform a comparative analysis. We have now added the simulation to the main text, together with a methodological explanation of how we simulated the data in the methods section. We also added a discussion on the results and why we think a hybrid approach is currently superior to synthetic approach. The whole new section is too long to paste here – it is found after the main simulation section in the manuscript. We have also added another sentence to the abstract referring to this new inclusion.

Finally, I think the authors could do further work to determine whether some of their recommendations for improving the sensitivity of TDLM pan out in the current data - for example, they could report focusing not just on the peak decoding timepoint but incorporating other moments into classifier training.

While we do understand the desire to test further refinement to TDLM on the data directly, we intentionally do not include such analyses in the current paper. Our experience also informs us that there is an enormous branching factor of parameters when applying TDLM, with implications for significance of results in one or other direction. However, as there are currently only limited ways to know how well parameter changes actually improve the sensitivity to replay versus exacerbate potential underlying confounders that induce spurious sequenceness (e.g., we can get significant replay in the control condition with some parameter changes). To exclude such false positive findings, we opt for a relatively strict adherence to previously published approaches. Thus, in the current paper, we limit ourselves to assessing the reliability and robustness of previous approaches.

Furthermore, while training on a later timepoint might increase sensitivity for a classifier when transferring between different modalities (e.g. visual to memory representation), this approach does not transfer well in our simulations, as the inserted patterns are from the same modality. We consider other, more bespoke studies, are better suited to improve classifier training. NB also see our recently started Kaggle challenge to tackle this problem: here.

However, we have added a note about this dilemma to the improvement section. The section now includes:

“Nevertheless, as the considerable branching factor poses a threat of increased falsepositive findings we opt to focus the current simulations on previously published pipelines and parameters. Future studies should systematically evaluate parameter choices on TDLM under different conditions, something that is beyond the remit of the current study.”

Lastly, I would like the authors to address a point that was raised in a separate public forum by an author of the TDLM method, which is that when replays "happen during rest, they are not uniform or close." Because the simulations in this work assume regularly occurring replay events, I agree that this is an important limitation that should be incorporated into alternative simulations to ensure the lack of findings is not because of this assumption.

The temporal distribution of replay throughout the resting state should not matter, as TDLM is invariant w.r.t to how replay events are distributed within the analysis window. Specifically, it does not matter if replay events occur in bursts or are uniformly distributed. Only the number of transitions is relevant, where they occur or if they are close to each other is not relevant to the numerical results (as long as the refractory window is kept, too short distances will lead to interactions between events and reduce sensitivity). To emphasize this point, we have added another simulation which is shown in Appendix A.1 and Appendix A Figure 1. We have referenced it in the text and added the following paragraph in the Methods section

Additionally, the timepoints of inserting replay within the resting state are sampled from a uniform distribution. Even though TDLM tracks reactivation events over time, at a macro-scale the algorithm is invariant to the temporal distribution. At each time step, the GLM regresses onto a future time step up to the maximum time lag of interest, yielding a predictor per lag. However, these predictors within the GLM are independently assessed, and hence, TDLM is, outside of the time lag window, relatively invariant to the temporal distribution of replay. To demonstrate our claim, we simulated uniform replay vs “bursty” replay that only occurs in some parts of the resting state, both yield equivalent sequenceness results (see Appendix A.1).

**Reviewer #3 (Public review):**
(1) I am still left wondering why other studies were able to detect replay using this method. My takeaway from this paper is that large time windows lead to high significance thresholds/required replay density, making it extremely challenging to detect replay at physiological levels during resting periods. While it is true that some previous studies applying TDLM used smaller time windows (e.g., Kern's previous paper detected replay in 1500ms windows), others, including Liu et al. (2019), successfully detected replay during a 5-minute resting period. Why do the authors believe others have nevertheless been able to detect replay during multi-minute time windows?

(Due to similarity, we combined our responses with the first question of Reviewer 1)

We are reluctant to make sweeping judgments in relation to previous literature as we wanted to prioritize on advancing methodology instead. The previous TDLM literature uses a diverse set of tasks and cognitive processes. As we state ourselves, it is possible that replay bursts in short time windows are well detectable by TDLM. We were intentionally cautious to directly critique previous studies without detailed re-analysis of their work and wanted to leave such a conclusion up to the reader. However, we realize that such a “thought-starter” might be warranted and improve the paper. Therefore, we have added the following paragraph to the discussion about “improving TDLMs sensitivity”:

“Finally, what do our simulations imply for the broader MEG replay literature? Our implementation successfully detects replay when boundary conditions are met, as shown in the simulation. But sensitivity depends critically on high fidelity between the analysis window and the amount of replay events. A systematic evaluation of these conditions across prior studies is beyond the scope of this paper, so we do not want to adjudicate earlier findings and leave this assessment up to the reader. Instead, we delineate the boundary conditions and urge future work to conduct power analyses where possible and include simulations that approximate realistic experimental conditions.”

For example, some studies using TDLM report evidence of sequenceness as a contrast between evidence of forwards (f) versus backwards (b) sequenceness; sequenceness was defined as ZfΔt - ZbΔt (where Z refers to the sequence alignment coefficient for a transition matrix at a specific time lag). This use case is not discussed in the present paper, despite its prevalence in the literature. If the same logic were applied to the data in this study, would significant sequenceness have been uncovered? Whether it would or not, I believe this point is important for understanding methodological differences between this paper and others.

This approach was first introduced as part of a TDLM-predecessor that utilized crosscorrelations (Kurth-Nelson 2016), where this step is a necessity to extract any sequenceness signal at all by subtracting signals that are present in both (akin to an EEG reference). However, its validity is less clear when fwd and bkw are estimated separately, as is in the GLM case. The rationale behind subtracting here is the same as for autocorrelations: there are oscillatory confounds present in the data that introduce spurious sequenceness in both directions alike, i.e. at the same time lag, that can simply be removed by subtracting. However, this assumption only holds if the sole confounder is auto-correlations caused by a global signal that oscillates at all sensors at the same phase. In our own experience, and mentioned in the discussion, we do not think this assumption holds. Arguably, there are more complex interactions at play that cannot be removed by such a subtraction such as an increase in false positives if confounders are in an opposite direction at a specific time lag. This assumption-violation can be seen in our baseline condition, where other spurious sequenceness diverges in opposite directions for some time lags (e.g. at ~90 ms where forward sequenceness is negative and backward sequenceness is positive). We reasoned that oscillatory confounds are more stable when comparing pre vs post for the same direction than comparing within session between forward minus backward.

Finally, we note issues introduced by the various ways that sequenceness has been analysed in previous papers: normalization of sequenceness (z-scoring across time lags or across participants or not at all), normalization of probabilities (taking raw decision scores, z-scoring, soft-max, dividing by mean, subtracting mean), taking a windowed approach and summing sequenceness scores, not to mention the various classifier choices that can be made, and all of this can be applied before subtracting conditions from each other or before subtraction. In our experience there is insufficient regard to control for multiple comparison when running all these analyses risking selectivity in reporting.

Nevertheless, subtracting forward from backward replay is probably as valid as post minus pre. Therefore, we have added fwd-bkw plots to the supplement and explained some of the reasoning for not reporting them in the main text in the figure label. The figure label and reference now read:

“Finally, we report forward minus backward sequenceness and our motivation for using an across-session post-pre comparison instead of within-session forwardbackward in Supplement Figure 10.”

[…]

“Forward minus backward sequenceness within each resting state session. Previous papers often report subtraction of backward from forward sequenceness (fwd-bkw) as a means to remove oscillatory confounds that impact both sequenceness directions in synchrony. While required in early cross-correlation approaches (KurthNelson et al., 2016), its validity in GLM-based frameworks depends on an assumption that confounds are global and in-phase across sensors. We observed this assumption is violated in our baseline data, where spurious sequenceness occasionally diverges in opposite directions at specific time lags (e.g., ~90 ms). In such instances, subtraction would increase the false-positive rate rather than suppress noise. In Figure 3B, we prioritized the comparison of pre-task versus post-task sequenceness within the same direction, as oscillatory confounds appeared more stable across time within a single direction, as opposed to across directions within a single session. However, we consider both approaches are valid. We now provide the fwd-bkw plots for completeness and comparison with previous literature. (A) forward minus backwards sequenceness for Control (left) and Post-Learning resting-state (right). (B) T-value distribution of the sign-flip permutation test for Control (left) and Post-Learning resting-state (right)”

(2) Relatedly, while the authors note that smaller time windows are necessary for TDLM to succeed, a more precise description of the appropriate window size would greatly improve the utility of this paper. As it stands, the discussion feels incomplete without this information, as providing explicit guidance on optimal window sizes would help future researchers apply TDLM effectively. Under what window size range can physiological levels of replay actually be detected using TDLM? Or, is there some scaling factor that should be considered, in terms of window size and significance threshold/replay density? If the authors are unable to provide a concrete recommendation, they could add information about time windows used in previous studies (perhaps, is 1500ms as used in their previous paper a good recommendation?).

We currently do not have an empirical estimate of which window sizes are appropriate. While we used 1500ms in our previous paper, this was solely given by the experiment design which had a 1.5s wait period before the next stimulus. Our recommendation for best guidance on this matter would be to investigate related intracranial literature for SWR rate increases under similar experimental conditions. We have added the following paragraph to the discussion:

“At this stage we cannot offer a general recommendation for window sizes as they are likely to depend on details of the research paradigm. However, intracranial recordings can be used as proxy to estimate the duration of replay bursts, for example as reported in (Norman et al., 2019) where increased SWRs were seen up to 1500 ms after retrieval cue onset”

(3) In their simulation, the authors define a replay event as a single transition from one item to another (example: A to B). However, in rodents, replay often traverses more than a single transition (example: A to B to C, even to D and E). Observing multistep sequences increases confidence that true replay is present. How does sequence length impact the authors' conclusions? Similarly, can the authors comment on how the length of the inserted events impacts TDLM sensitivity, if at all?

Good point! So far, most papers do not seem to include multi-step TDLM and in our experience rightfully, as it is conceptionally difficult to define clear significance thresholds while keeping in mind that shorter sub-sequences are contained within a longer sequence (e.g. ABC contains both AB and BC and a longer dependency of AC) that renders it difficult to define the correct way to create a null distribution for the permutation test. Therefore, we tried to stay as close as possible to previous approaches and only looked for single-step transitions. Nevertheless, we have added an analysis to the supplement comparing how TDLM behaves if we simulate A->B->C or A->B and separate B->C. It shows that TDLM is only sensitive to the number of transitions present in the data, and it does not matter if they are chained or chunked. The segment reads:

“We intentionally designed our study to encourage replay of triplets. However, this begs the question as to whether it matters if triplets or individual chunks of a sequence are replayed at different time points? Here, we simulated two scenarios. In one, we inserted replay of single transitions alone with a refractory period, e.g. A->B and separate B->C transitions. In a second scenario, we simulate replay of chained triplets, e.g. A->B->C, with a distance of 80 milliseconds each. Importantly, we kept the number of transitions constant i.e., A->B, … B->C and where A->B->C would both have 2 transitions. This creates a context wherein a four-minute resting state would have ~100 events of A->B->C inserted and ~200 events of A->B or B->C, such that in both cases this results in the same number of single step transitions. We found both are equivalent, with TDLM agnostic to the length of sequence trains, i.e., it does not matter if replay is chunked or chained under the assumption that the number of transitions remains fixed, as can be seen in Appendix A Figure 2”

And the reference Figure description reads:

“TDLM is invariant to the length of sequence replay trains under an assumption that the number of target transitions (e.g. single steps) is fixed. We simulated replay either as two temporally separate A->B, B->C events (light orange/green) or as a single A>B->C event (dark orange/green), both yielding equivalent sequenceness. Shaded areas denote the standard error across participants”

For example, regarding sequence length, is it possible that TDLM would detect multiple parts of a longer sequence independently, meaning that the high density needed to detect replay is actually not quite so dense? (example: if 20 four-step sequences (A to B to C to D to E) were sampled by TDLM such that it recorded each transition separately, that would lead to a density of 80 events/min).

Indeed, this is an interesting proposal. We intentionally kept our simulation close to the way previous simulations were set-up (i.e. Liu & Dolan et al 2021, Liu & Mattar 2021) by simulating one-step transitions and simulated them such that there is no overlap between separate events (e.g. by defining a refractory period). If the duration of replay is increased then we would also need to increase the length of the refractory period, resulting in a reduced upper limit of how much replay can occur in a 1-minute time window. This in turn would approximate roughly the same number of transitions that can be inserted into the resting state and, as detailed above, would yield the same results. Nevertheless, as we chose to use replay density and not transition density as a marker, the density would be reduced, even if the number of transitions stay the same. We have added an analysis using multi-step replay to the supplement and discuss its implications and caveats. In the main discussion we have added the following segment:

“Similarly, in our simulation, for simplicity and to keep consistency with previousstimulations, we restricted replay events to span two reactivation events. While the characteristics of replay as measured by TDLM are unknown, it is conceivable that several steps can be replayed within one replay event. We show that the vanilla version of TDLM is fundamentally sensitive to the number of single-step transitions alone, and disregards if these are replayed chained or chunked (Appendix A.2 and Appendix A Figure 2). Nevertheless, if the number of reactivation events chained within a replay event increases, TDLMs sensitivity is increased relative to the replay density and thresholds are reached earlier (see Appendix A Figure 4). See Appendix A.4 for a simulation of multi-step replay events and our discussion of the caveats.”

**Recommendations for the authors:**

**Reviewer #1 (Recommendations for the authors):**
Please label the various significance thresholds in the legend of Figure 3.

We have labelled all the thresholds in the figure legends.

**Reviewer #2 (Recommendations for the authors):**
I think that some of the clarity is hampered because there is a bit too much reliance on explanations from the previous paper using this task, which hampers clarity in the paper. For example, Figure 1 is not particularly useful for understanding the study in its current form; I found myself relying almost exclusively on Supplementary Figure 1 (which is from the previous paper). I'd recommend presenting some version of SF1 in the main text instead. Another example of this overreliance on the previous paper is that, as far as I can tell, the present paper never explicitly states which transitions are being tested in TDLM. In the prior work, it states "all allowable graph transitions", and so I assumed this was the same here, but the paper should standalone without having to go back to the other study. I'd recommend that the authors revise the paper in these and other places where the previous paper is mentioned.

Thanks for raising this point! We were uncertain ourselves how to deal with the overlap in content and did not want to bloat the paper or plagiarize ourselves too much. On the advice of the referee have implemented the following to improve the manuscript and reduce a reliance on the previous paper:

Supplement Figure 1 is indeed crucial to understanding the experiment. We have moved it to the methods section under Methods: Procedure

Added more stimulus description to the Methods: Localizer section

Included more details about the localizer and graph learning that were missing before

We have added the note about which transitions we were looking for in the Methods section. Additionally, we have added this information to the Results section of Study 1.

There are also a few typos I noticed:(1) Line 73: "during in the context of."(2) Line 287: " to exploring the."

We fixed the typos.

**Reviewer #3 (Recommendations for the authors):**
(1) Why did the authors choose an 80ms state-to-state time lag for their simulation? I believe they should make the reason for this decision clear in the main text.

Indeed, this point was also raised by the other reviewer. We have added a sentence to the main text about the rationale behind this decision:

“We chose this timepoint (80 millisecond state-to-state lag) as its sequenceness value was close to zero in the baseline condition as well as being distant to the observed alpha rhythms of the participants (which varied between ~9-11 Hz). Additionally, we subtracted the mean sequenceness value of the baseline at 80 millisecond lag such that any simulation effects would, on average, start at zero sequenceness.“

Additionally, we have added some further explanation to the Methods section.

“This time lag (80 msec) was chosen in order to isolate precisely an effect of the experimentally inserted sequenceness. Thus, we chose a lag at which the mean baseline sequenceness was close to zero and where the correlation with behaviour was low. Additionally, we subtracted the mean sequenceness value (at 80 milliseconds) at baseline from the specific lag recorded for each participant, such that simulation effects would be initialized at zero sequenceness on average enabling any effects to be attributed purely to inserted replay. Additionally, we excluded time lags too close to the alpha rhythms of participants (which varied between ~9-11 Hz) or lags which would have a harmonic with the rhythm.“

(2) Line 168: Can the authors define what these conservative and liberal criteria are in the text?

We have added definitions of the criteria in the text. The text now reads:

“[..] significance thresholds (conservative, i.e. the maximum sequenceness across all permutations and timepoints or liberal criteria, i.e. the 95% percentile of aforementioned sequenceness).”

(3) Line 478: "calculate" instead of "calculated".(4) Figure 7 D: y-axis is labeled "70 ms" I believe it should be labeled 80 ms.

Thanks, we fixed the two typos.

(5) With replay defined as sequential reactivation at a compressed temporal timescale, many of the iEEG citations (lines 54-55) do not demonstrate replay (they show stimulus reinstatement or ripple activity, but not sequential replay). Replay studies in humans using intracranial methods have been mostly limited to those measuring single-unit activity, a good example being Vaz et al., 2020 (https://www.science.org/doi/10.1126/science.aba0672).

We agree that, under a strict definition articulated by Genzel et al. that defines replay as sequential reactivation, many prior human iEEG studies are better described as stimulus reinstatement or ripple-related activity rather than true sequence replay. We have revised the text accordingly and now highlight the few intracranial microelectrode studies that demonstrate replay of firing sequences at the cellular/ensemble level in humans (Eichenlaub et al., 2020; Vaz et al., 2020), distinguishing these from macro-scale iEEG work providing indirect evidence alone.

The revised paragraph now reads:

“Replay has been shown using cellular recordings across a variety of mammalian model organisms (Hoffman & McNaughton, 2002; Lee & Wilson, 2002; Pavlides & Winson, 1989). Replay studies in humans using intracranial recordings are few, but include work demonstrating compressed replay of firing-pattern sequences in motor cortex during rest (Eichenlaub et al., 2020) as well as single-unit replay of trialspecific cortical spiking sequences during episodic retrieval (Vaz et al., 2020). By contrast, most iEEG studies report stimulus-specific reinstatement or ripple-locked activity changes without explicit demonstration of temporally compressed sequential replay (Axmacher et al., 2008; Staresina et al., 2015). As these methods are only applied under restricted clinical circumstances, such as during pre-operative neurosurgical assessments, this limits opportunities to investigate human replay. Therefore, this gives urgency to efforts aimed at developing novel methods to investigate human replay non-invasively.”

(6) The expectations about replay frequency are grounded in literature on hippocampal replay sequences. However, MEG captures signals from across the entire brain, and the hippocampal contribution is likely relatively weak compared to all other signals. This raises an important question: is TDLM genuinely unable to detect replay at physiological (i.e., hippocampal) levels, or is it instead detecting a different form of sequential reactivation - possibly involving cortex or other regions - that may occur more frequently? More broadly, when we have evidence of replay from TDLM, do we believe it is the same thing as replay of CA1 place cell spiking sequences, as detected in rodents? Commenting on this distinction would help further develop theories of replay and what TDLM is measuring.

This is indeed an important point that has garnered relatively little attention. While there is some evidence of a relation to hippocampal replay in form of high-frequency power increase in the hippocampus, ultimately it is not possible to know without intracranial recordings, as signal strength from those regions is rather poor in MEG.

We have added the following segment to the manuscript that discusses these issues:

“However, while we are using indices of SWRs as a proxy for replay density estimation, the relationship between hippocampal replay and replay detected by TDLM remains uncertain. While current decoding approaches measure replay-like phenomena on cortical sites, previous papers have reported a power increase in hippocampal areas coinciding with replay episodes as detected by TDLM. Nevertheless, it is conceivable that cortical replay found by TDLM could occur independently of hippocampal replay and SWRs and be generated by different mechanisms. Some TDLM-studies find a replay state-to-state time lag of above 100 ms, much slower than e.g. previously reported place cell replay. Future studies should employ simultaneous intracranial and cortical surface recordings to establish the relationship between hippocampal replay and replay found by TDLM.”